# An Efficient Subset Selection Strategy Using Text-Guided Data Attribution to Mitigate Simplicity Bias

**Kumar Shubham**                                          *shubhamkuma3@iisc.ac.in*
*Indian Institute of Science, Bangalore, India*

**Pranav Sastry**                                          *pranavsastry@iisc.ac.in*
*Indian Institute of Science, Bangalore, India*

**Prathosh AP**                                          *prathosh@iisc.ac.in*
*LatentForce.ai*
*Indian Institute of Science, Bangalore, India*

**Reviewed on OpenReview:** *https://openreview.net/forum?id=zZ5YundT95*

## Abstract

The effectiveness of deep learning models heavily relies on the quality and diversity of their training data. However, datasets collected from different sources often introduce simplicity biases, where a models rely on easily learnable but non-predictive (spurious) features for its predictions. While existing debiasing techniques focus on model robustness, they leave the data untouched. However, as data becomes increasingly valuable, identifying and mitigating bias directly at the data level has become increasingly important. Recently, data attribution has emerged as a promising tool for uncovering issues in training data, yet its vulnerability to simplicity bias has received limited attention. In this work, we propose a novel data deletion framework that combines Neural Tangent Kernel (NTK)-based data attribution with textual descriptions of bias to identify and remove training samples that do not significantly affect model performance. We first demonstrate that NTK-based data attribution methods can themselves be influenced by spurious features. Subsequently, to mitigate this, we use available metadata or, when unavailable, a vision-language model, to annotate a small validation set and extract a textual description of the bias. Based on this description and the attribution score, we identify the subset of training data that are semantically aligned with the spurious feature and affect the generalization of the model. Removing these samples from the training dataset and training model on the new subset improves the average and worst-group accuracy of the model, outperforming existing attribution-based baselines. Our code is available at https://github.com/kyrs/attribute-guided-subset-selection.

## 1 Introduction

The success of deep learning models is strongly influenced by the quality and quantity of the dataset used for training (Bhatt et al., 2024; Whang et al., 2023; Xu et al., 2024a; Choe et al., 2024). These data are often collected via web scraping (Xu et al., 2024b; Patel & Patel, 2020), and external data providers (Berriel et al., 2017; Drutsa et al., 2019). However, such datasets can inadvertently contain illegal content (Thiel, 2023) and can encode negative societal biases (Ferrara, 2023; Jain et al., 2024) that can influence model performance. In addition, data collected from such varied sources can introduce distributional shifts, where subpopulations with specific features may be overrepresented or underrepresented in the training data compared to the test data (Koh et al., 2021). For instance, in a traffic sign classification task, a significant portion of the training data collected online may consist of images captured on sunny days, while the model may later be deployed in winter or rainy conditions, leading to performance degradation.

These imbalances can introduce simplicity bias (Tiwari & Shenoy, 2023; Geirhos et al., 2020; McCoy, 2019) where the model, due to high correlations between specific features and the prediction task, relies on simpler, non-robust (spurious) features instead of learning predictive features for classification. Several methods have been proposed to mitigate such biases. However, most of these approaches primarily focus on enhancing model robustness by modifying the training process (Liu et al., 2021; Sagawa et al., 2020; Lee et al., 2021; Liu et al., 2022; LaBonte et al., 2023) and leverage the same training dataset to improve performance across various settings. Since these methods operate directly on the dataset from which the biases originate, they may fail to fully address underlying distributional disparities. Moreover, in certain scenarios, these changes in the training process can inadvertently affect the model's susceptibility to adversarial attacks (Hartnett et al., 2019; Li et al., 2025; Neerudu et al., 2023; Holtz et al., 2022), and may conflict with regulatory constraints (Wang et al., 2018; Wong & Kolter, 2018) that mandate specific training protocols to preserve theoretical guarantees (see Appendix J for further details). Further, given the proprietary value of data (Xiong et al., 2022) and the significant financial cost associated with curating new datasets, it has become increasingly important to address these challenges directly at the data level.

A viable alternative in these scenarios could be to remove training samples associated with spurious features (Chaudhuri et al., 2023; Idrissi et al., 2022; Jain et al., 2024), while ensuring that removal of such samples doesn't hurt the overall performance of the model, as in data attribution and Leave-One-Out (LOO) techniques (Koh & Liang, 2017; Tanno et al., 2022; Park et al., 2023). Data attribution methods aim to estimate a model's performance when specific training samples are excluded, enabling the evaluation of counterfactual scenarios—such as assessing the impact on test accuracy if certain subsets of the training data were omitted (Tanno et al., 2022; Engstrom et al., 2024; Xia et al., 2024). However, many of these methods are computationally expensive and can underperform for a non-convex training objective. Recent advancements in data attribution methods, such as Trak (Park et al., 2023), leverage neural tangent kernels (NTK) to enable scalable data attribution for non-convex models (Park et al., 2023). However, the impact of spurious features on the data attribution scores generated by such methods remains an open question.

In our work, we demonstrate (Proposition 1, Section 4.8) that in the presence of data bias, methods like Trak (Park et al., 2023) can undervalue the attribution scores for training samples with spurious features (Tiwari & Shenoy, 2023; Geirhos et al., 2020; McCoy, 2019). This misattribution can hinder the identification of detrimental samples, especially for methods that rely solely on the magnitude of attribution scores (Marion et al., 2023; Tanno et al., 2022).

Motivated by these observations, we propose a two-stage strategy to mitigate the impact of spurious features - (a) In the first stage, we focus on identifying such features within the dataset using available meta-data or annotations generated by a vision language model. (b) In the second stage, we use multimodal embeddings, such as CLIP (Radford et al., 2021) to learn a metric (Lim & Lanckriet, 2014; Bhalla et al., 2024) that identifies training examples that are semantically similar to the spurious features identified in the first step and whose removal can improve the model's performance as per the attribution scores.

The spurious features in the first stage are identified using metadata wherever available. In cases where metadata is unavailable, we utilize a vision-language model (VLM) to annotate a small validation set with its respective attributes and their associated values that are likely to introduce simplicity biases (Chen et al., 2024; Lu & Zhong, 2024; Tan et al., 2024). By evaluating the model's performance on these attribute-value pairs and comparing it to the overall performance on the validation dataset (Johnson et al., 2023), we identify potential spurious features and generate a corresponding textual description of these biases (Eyuboglu et al., 2022). This textual representation enables targeted data pruning and helps to mitigate the impact of spurious features without relying on manual group annotations in the training dataset.

In summary, our contributions in this paper are as follows:

- We propose a novel data-centric approach that combines NTK-based data attribution methods with textual descriptions of underlying bias to mitigate the impact of spurious features in training datasets.

- We first theoretically demonstrate that NTK-based attribution scores can be influenced by spurious features, which may limit the effectiveness of methods that rely solely on these scores for data

pruning. To overcome this limitation, we introduce a metric learning-based data deletion strategy that selectively removes training samples aligned with textual descriptions of spurious features and with a high detrimental attribution score.

- Our approach achieves up to a 4% gain in average accuracy, 18% in worst-group accuracy, and a 50% improvement in class-level performance across various datasets. Additionally, it outperforms NTK-based methods like Trak on average by 10.6% in worst-group accuracy for different biased datasets.

## 2 Related Work

### 2.1 Data Attribution

Data attribution methods provide a framework to relate a model's predictions to its training dataset and have been used in a wide range of tasks, including model debugging and repair (Yeh et al., 2018; Tang et al., 2021; Shah et al., 2023; Grosse et al., 2023), subset selection (Engstrom et al., 2024; Xia et al., 2024; Chhabra et al., 2024), group robustness (Jain et al., 2024), and removing poisoning attacks (Wu et al., 2023a).

The idea of linking a model's predictions to its training data has been studied for decades under various names, including influence functions (Hampel, 1974), regression analysis (Pregibon, 1981), and jackknife methods (Miller, 1974). However, most of these early works focused on linear models and aimed to predict changes in the optimal parameters when individual or groups of samples were excluded during the learning process. Recent works have tried to extend influence function and jacknife-based attribution methods to non-linear models and bigger datasets (Koh & Liang, 2017; Rad & Maleki, 2018; Giordano et al., 2019). However, despite their promising predictive capabilities, these methods often make strong assumptions of convexity and the existence of a unique global solution, which are not applicable for neural networks (Bae et al., 2022). Furthermore, Basu et al. (2020); Hammoudeh & Lowd (2022) have demonstrated the fragile nature of methods like influence functions across different architectures, showing that they sometimes fail basic sanity checks. Various approaches have been proposed to address the limitations of influence functions, including gradient agreement scoring (Pruthi et al., 2020), training models to predict attribution scores, as in DataModels (Ilyas et al., 2023; Engstrom et al., 2024), and methods like Trak (Park et al., 2023), which leverage concepts from the Neural Tangent Kernel (NTK) for data attribution. Unlike other approaches, such as DataModels, Trak does not require training thousands of models (Park et al., 2023; Ilyas et al., 2023) or tracking the loss changes over the entire training process, making it more efficient. However, the impact of spurious features within the dataset on the data attribution method like Trak remains largely unexplored.

### 2.2 Spurious Features and Simplicity Bias

Spurious features often arise from selection bias in the dataset (Ye et al., 2024), where, in the presence of multiple hypotheses for prediction, the model tends to rely on the simplest feature (Pezeshki et al., 2021; Geirhos et al., 2020; Tiwari & Shenoy, 2023). This preference can lead to suboptimal model performance, as it often ignores more robust and meaningful features that are essential for generalization in real-world scenarios. Various methods have been proposed to address spurious features in models. These include data augmentation techniques (Srivastava et al., 2020; Puli et al., 2022; Yao et al., 2022; Zeng et al., 2020; Wu et al., 2023b; Nam et al., 2022; Zarlenga et al., 2024), and learning strategies that change the training objectives to make the model robust to spurious features (Sagawa et al., 2020; Wang et al., 2021; Levy et al., 2020; Liu et al., 2022; 2021; Kirichenko et al., 2023; LaBonte et al., 2023; Han & Zou, 2024). However, many of these changes are restricted under the regulatory policy for safety-critical applications (Petersen et al., 2022; Matheny et al., 2019; Shen et al., 2023; Campos Zabala, 2023), especially considering privacy concerns associated with collecting datasets and model certification-based requirements (Valentin, 2024; Liu et al., 2024). Recent works have explored data deletion as a strategy for mitigating spurious features (Chaudhuri et al., 2023; Idrissi et al., 2022; Jain et al., 2024). These methods use group annotation of the dataset to remove random samples from majority groups (Chaudhuri et al., 2023; Idrissi et al., 2022) or those with high detrimental attribution scores (Jain et al., 2024). However, these methods often require manual group annotation of training (Chaudhuri et al., 2023; Idrissi et al., 2022) or validation data (Jain et al., 2024),

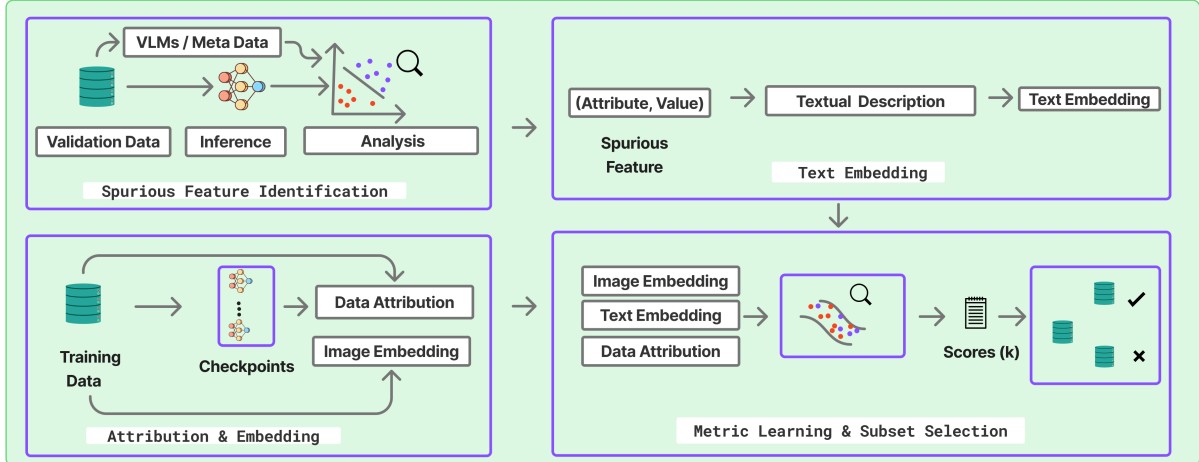

Figure 1: The figure illustrates the key steps in identifying detrimental samples. First, the performance of the model across different attribute value pairs is analyzed to identify and textually describe the underlying bias. Then, training samples that align with this bias and exhibit high detrimental attribution scores are selected for removal.

which is costly and time-consuming. Further, in real-world settings, where biases are identified post hoc after deployment and evolve over time (Lesort, 2023), generating such annotations is often impractical, and enforcing a balance among different groups may result in excessive data removal from the majority group and can harm generalization (Idrissi et al., 2022). Our method circumvents these limitations by using text-guided data attribution to efficiently remove harmful samples within a deletion budget, without relying on group labels or hurting model performance. Further details on limitations and capabilities of existing methods are discussed in Appendix J.

## 3 Proposed Method

### 3.1 Problem Definition

Consider a classification setting with a training dataset $\mathcal{D}_{\text{train}} = \{z_1, \ldots, z_n\}$, where each sample $z_i = (x_i, y_i)$, consists of an input $(x_i)$ and associated class label $(y_i)$ and an unbiased validation dataset, $\mathcal{D}_{\text{val}} = \{v_1, \ldots, v_m\}$ with validation samples $v_j = (x_j, y_j)$. The training dataset $(\mathcal{D}_{\text{train}})$ is used to train a neural network with optimal parameters $\theta^*(\mathcal{D}_{\text{train}})$. Additionally, we assume that $|\mathcal{D}_{\text{val}}| \ll |\mathcal{D}_{\text{train}}|$.

Suppose for every training sample $z$ there exists $\boldsymbol{t}$ underlying hidden discrete attributes, $A' = \{a^1, \ldots, a^{\boldsymbol{t}}\}$ and for each attribute $(a^j)$ there are $\boldsymbol{o}$ possible values denoted as $V(a^j) \in \{b_1^j \ldots b_{\boldsymbol{o}}^j\}$. In real-world settings, neural networks $(\theta^*)$ trained on $\mathcal{D}_{\text{train}}$ often associate class labels $(y)$ with specific attribute-value pairs $(a^m, b_t^m)$ (Eyuboglu et al., 2022; Tiwari & Shenoy, 2023; Geirhos et al., 2020). For example, a model trained to predict gender might associate it with the feature "beard" (present/absent). However, feature imbalance in the datasets can lead to misleading associations. If most of the male images in a dataset include smiles, the model might spuriously link "male" with "smiling" rather than "beard." This can cause misclassification, like predicting smiling females as males. We term such misleading attribute-value pairs as **spurious features**. In these scenarios, the primary objective of our work is to identify a set of detrimental examples, $\mathcal{S}^{\text{deter}} \subset \mathcal{D}_{train}$, with features similar to the spurious features and whose removal does not degrade the model's performance. Training the model on the filtered subset $(\mathcal{D}_{\text{train}} \setminus \mathcal{S}^{\text{deter}})$ improves its performance compared to the original dataset, and reduces the influence of the spurious feature in the training dataset, similar to prior work like Chaudhuri et al. (2023). Further details on training and validation dataset are provided in Apeendix I.6.

Our method for identifying $\mathcal{S}^{\text{deter}}$ involves two steps: (1) Annotate attribute–value pairs in the validation set to detect potential spurious features and generate a textual description of the bias; (2) Select $\mathcal{S}^{\text{deter}} \subset \mathcal{D}_{\text{train}}$ as samples semantically aligned with the bias and whose removal as per the data attribution scores does not degrade model performance.

## 3.2 Attribute Annotation and Spurious Feature Identification

A key component to identify spurious features is the availability of attribute–value annotations for the validation dataset. However, in many practical scenarios, such annotations are often missing from the metadata. Chen et al. (2024) has shown that in the absence of such information, large language and vision models can be used to generate annotations necessary to identify the underlying spurious features. Hence, for datasets without pre-annotated attributes, we annotate the validation set with potential attribute–value pairs to assist in identifying spurious features.

To generate candidate attribute–value pairs, we leverage LLM-based tool such as ChatGPT (Chen et al., 2024). ChatGPT is provided with a simple task description and prompted to suggest relevant attributes and associated values. For example, in a classification task of distinguishing between land birds and water birds, ChatGPT can generate attributes such as "habitat type" or "background environment," with values like "forest" for land birds and "lake" for water birds. We adopt the task-specific prompts proposed by Chen et al. (2024) to guide this process.

Once the attribute–value pairs are generated, the next step is to annotate the validation dataset. While ChatGPT excels at generating textual descriptions, it is not suited for image annotation (Chen et al., 2024). To address this, we use Llama 3.2 (Dubey et al., 2024), a vision–language model, to annotate a small set of validation images with the corresponding attribute–value pairs (Chen et al., 2024). Further details about the prompts can be found in Appendix H.

### 3.2.1 Spurious Feature Identification

To identify spurious features, we take motivation from recent work that tries to identify systematic bias in a model (Johnson et al., 2023; Eyuboglu et al., 2022) based on its accuracy and errors on the unbiased validation dataset. However, unlike previous methods, which try to identify underperforming subgroups that may require collecting additional data, we try to determine the overperforming attribute-value pair as a possible candidate for data deletion (Chiu et al., 2023; Chaudhuri et al., 2023). For this analysis, we take inspiration from Johnson et al. (2023) and compare the performance of the dataset corresponding to each attribute–value pair with the performance of the overall dataset. In general, robust and informative features tend to improve both the subgroup accuracy (associated with a given attribute–value pair) and the average accuracy across the dataset. Conversely, a large performance gap between these two can indicate that improvements for a particular subgroup come at the expense of others (Johnson et al., 2023; Eyuboglu et al., 2022). In our framework, if this performance gap exceeds a predefined threshold, the corresponding attribute–value pair is flagged as a potential spurious feature learned by the model. Formally, this is expressed as:

$$\frac{1}{\left|\mathcal{D}_\alpha\right|} \sum_{(x,y)\in\mathcal{D}_\alpha} \mathbf{1}\big(h(x)=y\big) - \frac{1}{\left|\mathcal{D}_{\text{val}}\right|} \sum_{(x,y)\in\mathcal{D}_{\text{val}}} \mathbf{1}\big(h(x)=y\big) > \tau, \tag{1}$$

where, $\mathcal{D}_\alpha$ is a subset of validation data $\mathcal{D}_{val}$ associated with $a^v$ attribute and it's $j^{th}$ value $b_j^v$. The indicator function $\mathbf{1}$ indicates the correct prediction made by the model. The function $h(x)$ represents the prediction made by the model for a given input x ; and y is the corresponding true class label. The parameter $\tau$ denotes the minimum threshold.

Once an attribute-value pair exceeds the threshold, a textual description is generated to describe the spurious feature. For example: *"Images with {a} as {b}."* Here, $(a, b)$ is the attribute value pair selected as per Equation 1. Details about the textual description and hyperparameters are provided in Appendix I.5 and Appendix I.2, respectively.

### 3.3 Coherent Data Attribution

After generating the desired text, the next task is to select a subset of data that is semantically coherent with the given text and whose removal does not degrade the performance of the model (Huang et al., 2024).

Since our task involves efficient subset selection, we formally define data attribution as follows:

**Definition 1** (Data Attribution and Leave-one-out Influence Score (Park et al., 2023)). *Given training dataset $\mathcal{D}_{train}$, and a model's utility function $\boldsymbol{f}(v; \theta)$ that measures the performance of the model, the data attribution score $\alpha : \mathcal{D}_{train} \times \mathcal{D}_{val} \to \mathcal{R}$ is defined as the change in the model's prediction for a validation sample $v_i$ with respect to the optimal parameters when the training example $z_k$ is excluded from the training dataset during the learning of the optimal parameters $\theta^*$. Formally,*

$$\boldsymbol{\alpha}(v_i; z_k) = \boldsymbol{f}\left(v_i; \theta^*(\mathcal{D}_{train})\right) - \boldsymbol{f}\left(v_i; \theta^*(\mathcal{D}_{train} \backslash z_k)\right) \tag{2}$$

For a classification task, the utility function $\boldsymbol{f}(z; \theta)$ for a sample $z = (x, y)$, (Park et al., 2023), is defined as:

$$\boldsymbol{f}(z; \theta) = \log\left(\frac{p(z; \theta)}{1 - p(z; \theta)}\right), \tag{3}$$

where $p(z; \theta)$ represents the probability assigned to the correct class by the softmax function of a neural network parameterized by $\theta$. A high $\boldsymbol{f}(z; \theta)$ corresponds to a high likelihood for a given sample ($z$).

The NTK-based methods like Trak have a closed-form formulation for data attribution score ($\alpha$) (Definition 1) expressed as:

$$\boldsymbol{\alpha}(v_j, z_i) = \frac{1}{N} \sum_{n=1}^{N} \left(\phi_n(v_j)^\top (\Phi_n^\top \Phi_n)^{-1} \phi_n(z_i)\right) \times \frac{1}{N} \sum_{n=1}^{N} \left(1 - p_n^{z_i}\right) \tag{4}$$

where, for $N$ different checkpoints of model ($\theta^*$), $\{p_n^{z_i}\}$ represents the probability assigned by $n^{th}$ set of parameters to the correct class ($y_i$), for sample $z_i = (x_i, y_i)$. The terms $\phi_n(v_j)$ and $\phi_n(z_i)$ denote the projected gradients of the validation sample $v_j$ and the training sample $z_i$ with respect to the $n^{th}$ set of optimal parameters, and for the utility function $\left(\boldsymbol{f}(\cdot; \theta_n^*)\right)$. Additionally, $\Phi_n$ is the projected gradient for the entire training dataset. Further details about Trak can be found in Appendix B.

To quantify the impact of removing a data sample $z$ from the training dataset on the performance of the entire validation dataset, we define the metric $\mathcal{A}(z)$ as a detrimental attribution score associated with the validation dataset for sample $z$. This metric measures the change in the model's performance $\left(\boldsymbol{f}\right)$ for the entire validation dataset when $z$ is excluded from the training dataset.

$$\begin{aligned} \mathcal{A}(z_i) &= - \sum_{v_j \in \mathcal{D}_{\text{val}}} \boldsymbol{\alpha}(v_j, z_i) \\ &= \sum_{v_j \in \mathcal{D}_{\text{val}}} \left(\boldsymbol{f}(v_j; \theta^*(\mathcal{D}_{\text{train}} \setminus z_i)) - \boldsymbol{f}(v_j; \theta^*(\mathcal{D}_{\text{train}}))\right) \end{aligned} \tag{5}$$

where $z_i \in \mathcal{D}_{\text{train}}$. Unlike the data attribution score defined in Definition 1, $\mathcal{A}(z_i)$ is the negative of the general definition and evaluates the contribution of each training sample to the likelihood of the entire validation dataset. A higher value of $\mathcal{A}(z_i)$ indicates that removing the training sample $z_i$ and retraining the model with the updated dataset leads to an optimal parameter $\theta^*$ that improves the likelihood of the validation dataset (Equation 3). In other words, training examples that degrade overall validation performance are assigned higher $\mathcal{A}(z_i)$ values. Once $\mathcal{A}(z_i)$ is calculated, it is normalized and used for further steps.

While removing samples with high $\mathcal{A}(z)$ values can improve the model's performance; however, its impact on the downstream model is often tied up with its capability to remove samples with spurious features. During training, spurious features present in the dataset can result in gradient starvation (Tachet et al., 2018; Pezeshki et al., 2021), a phenomenon that can hamper the learning of predictive features. Under such

scenarios, we theoretically show that the detrimental attribution score($\mathcal{A}$) for a data sample containing a spurious feature ($f_1$) can be lower than that of a data sample with predictive features ($f_2$), even when both features are equally represented. Consequently, deletion strategies based solely on high attribution scores may inadvertently remove examples with predictive rather than spurious features (Proposition 1) and can fail to capture the impact of removing data associated with spurious features on the overall generalization.

**Proposition 1** (Illustration for the Under Valuation of Attribution Scores)**.** *Consider a neural network in the neural tangent kernel (NTK) regime, trained using binary cross-entropy loss and contain two important features $f_1$ and $f_2$. Suppose that due to learning dynamics, $f_1$ becomes dominant and causes gradient starvation of $f_2$ as per Pezeshki et al. (2021). Let $z_i$ and $z_j$ be two training samples for which $f_1$ and $f_2$ are the most informative features, respectively, and assume that the they contain equal representations of these features (equivalently, $u_1^i = u_2^j$ as per Definition 3). Under these conditions, the detrimental attribution score for $z_i$ can be systematically undervalued relative to $z_j$. Formally:*

$$\left| \mathcal{A}(z_i) \right| < \left| \mathcal{A}(z_j) \right|$$

*The proof of Proposition 1, along with further details on gradient starvation, is provided in Appendix E. Empirical evidence supporting this phenomenon is presented in the Experiment section (Section 4.8, Appendix S).*

This limitation of attribution scores motivates the need for a targeted removal strategy that specifically identifies and eliminates training samples sharing similar spurious features and exhibiting high $\mathcal{A}(z)$ scores. In many practical scenarios, the information about spurious features is missing in the data. Although annotating the entire training dataset using VLM-based models is possible, this approach is often excessively time-consuming and practically infeasible, particularly for large-scale datasets (Lu & Zhong, 2024). To address this, we adopt a zero-shot approach (Pan et al., 2024) and leverage textual descriptions of bias and CLIP embeddings to select data samples that are semantically similar to the identified textual descriptions. The use of CLIP embeddings further allows us to capture fine-grained association between the textual description of bias and the input dataset, which is often hard to capture by group annotation or describe textually (Eyuboglu et al., 2022). Specifically, we convert the textual description (Section 3.2) of the potential spurious feature into an embedding $\mathcal{C}_{\text{text}}$. Similarly, we convert all images in the training dataset into their corresponding CLIP embeddings $\mathcal{C}_{\text{image}}^i$ for $i \in 1, \ldots, |\mathcal{D}_{\text{train}}|$. Each training sample $z_i$ is then assigned a score $\boldsymbol{k}_i$, reflecting its semantic similarity to the identified bias as per the given equation :

$$\boldsymbol{k}_i = \exp\left( -\frac{\left(\mathcal{C}_{\text{text}} - \mathcal{C}_{\text{image}}^i\right) M \left(\mathcal{C}_{\text{text}} - \mathcal{C}_{\text{image}}^i\right)^\top}{2} \right),$$

$$\text{where, } M = LL^\top, L \in \mathbb{R}^{D \times t}, t \ll D \tag{6}$$

The text and image features, denoted as $\mathcal{C}_{\text{text}}, \mathcal{C}_{\text{image}}^i \in \mathbb{R}^{1 \times D}$ are represented as row vectors in a D-dimensional space. The matrix $M$ is a positive semi-definite matrix, constructed as the outer product of a low-rank matrix $L$ (rank at most $t$), and can serve as a learnable transformation. Since $M$ defines the distance metric, varying the values of $L$ allows us to generate different similarity measures for comparing data points (Lim & Lanckriet, 2014; Bhalla et al., 2024).

We aim to remove data samples that are detrimental to the model's generalization (i.e., those with high $\mathcal{A}$ scores) and are semantically aligned with the identified bias. To accomplish this, we learn the matrix $L$ (d'Eon et al., 2022; Lim & Lanckriet, 2014; Bhalla et al., 2024) by maximizing the weighted $\mathcal{A}$ score for each sample, where higher weights correspond to stronger semantic alignment with the bias, as defined in Equation 6.

The optimization objective prioritizes learning a metric $L$ that assigns higher $k_i$ values to samples that both (i) exhibit high detrimental scores and (ii) align with the semantic description of the bias. The trade-off between these two objectives is governed by the hyperparameter $\mathcal{T}$, which constrains the cumulative score across the dataset to exceed a threshold defined as a fraction ($\beta$) of the total training size ($|\mathcal{D}_{\text{train}}|$). A larger $\mathcal{T}$ emphasizes semantic alignment, as defined in Equation 6, whereas a smaller $\mathcal{T}$ allows greater flexibility in selecting samples based primarily on their $\mathcal{A}$ scores. The complete optimization objective is described below:

$$\max_L \sum_{i=1}^{|\mathcal{D}_{\text{train}}|} \left( \frac{k_i}{\sum_j k_j} \right) \mathcal{A}(z_i) \quad \text{s.t.} \quad \sum_{i=1}^{|\mathcal{D}_{\text{train}}|} k_i \geq \mathcal{T}, \quad \mathcal{T} = \beta \times |\mathcal{D}_{\text{train}}|. \tag{7}$$

To ensure that the optimization remains tractable, we replace the hard constraint with a soft penalty term (d'Eon et al., 2022) in the objective function. Further detail on this is provided in Appendix D.

Once the optimization is complete, a subset of training data with $k_i$ scores greater than the hyperparameter $\gamma$ is selected for removal ($\mathcal{S}^{\text{deter}}$). The model is then retrained with the updated training dataset ($\mathcal{D}_{\text{train}} \backslash \mathcal{S}^{\text{deter}}$) where, $\mathcal{S}^{\text{deter}} = \{z_i \in \mathcal{D}_{\text{train}} \mid k_i > \gamma\}$. A sensitivity analysis of all the hyperparameters and subset size is provided in Appendix O and Appendix P, respectively.

## 4 Experiments

### 4.1 Setting

We evaluate the performance of our method across various datasets and compare it with existing data attribution techniques, including original training of the model with complete dataset (original), Random deletion of data points (Random), Influence Function (IF) (Koh & Liang, 2017), TracIN (Pruthi et al., 2020), EWC Repair (Tanno et al., 2022), and Trak (Park et al., 2023). The datasets used in our experiments include WaterBirds (Sagawa* et al., 2020), Animal with attributes (AWA2) (Xian et al., 2018), German Traffic Sign Recognition Benchmark (GTSRB) (Stallkamp et al., 2012), CELEBA (Liu et al., 2015; Eyuboglu et al., 2022; Zhang et al., 2022) (Appendix I), CIFAR-10 (Krizhevsky et al., 2009), and ImageNet-100 (Russakovsky et al., 2015; Tian et al., 2020). Further comparisons with robustness-based methods(groupDRO (Sagawa et al., 2020), JTT (Liu et al., 2021)) and group balancing methods are provided in Section 4.9. For datasets such as GTSRB, CIFAR-10, and WaterBirds, we utilized attributes generated by ChatGPT and VLM models. To further assess the impact of metadata availability, we created two variants for the AWA2 datasets. The first variant, AWA2-A, includes class-specific annotations provided by the original datasets. The second variant, AWA2-B, uses attributes generated using ChatGPT and VLM-based annotation techniques (Section 3.2). All primary experiments were conducted using a ResNet-18 model, which is the base architecture used in NTK-based data attribution methods such as Trak (Park et al., 2023) for the image classification task. Additional experiments using alternative architectures and vision transformer models are presented in Appendix L and Appendix M, respectively. We have reported the worst group accuracy and average accuracy based on prior work on spurious features (Nam et al., 2022; Sagawa et al., 2020; Chaudhuri et al., 2023). However, due to the absence of well-defined group structures in many real-world datasets (Sagawa et al., 2020), we have compared these datasets on average accuracy and class-level accuracy. All the experiments were conducted on two NVIDIA A6000 GPUs. Further details on training, hyperparameters, and subset size are provided in Appendix I. Algorithm 1 (Appendix) illustrates the overall workflow of our approach. We also report time and memory overheads associated with subset selection in Appendix T and Appendix U, respectively. Sample images from the selected subset $\mathcal{S}^{\text{deter}}$ are shown in Appendix W.

### 4.2 Improvement in Average Accuracy

Table 1 reports the improvement in average accuracy achieved by our method compared to existing baselines. On average, our method outperforms Trak by 1.4%, EWC by 1.6%, TracIN by 1.4%, Influence Functions by 2.0%, and the original full-dataset training baseline by 1.7%. Notably, we observe gains of 1.9%, 2.5%, and 2.4% over Trak on AWA2-B, WaterBirds, and AWA2-A, respectively. The performance improvement highlights the efficiency of our method in removing the detrimental samples associated with spurious features. We further saw a substantial improvement in under represented class as discussed in Section 4.3. Additional experiments on worst-group accuracy and architectural ablations for WaterBirds are provided in Appendix L.

### 4.3 Class Level Improvement after Data Deletion

Table 2 presents class-level accuracy for datasets with more than two classes. As per the results, our method improves the accuracy of a significant number of classes across datasets. For example, in Awa2-A, Awa2-B, ImageNet-100, and CIFAR-10, over 40% of the classes show improvement, with some achieving gains as high as 29.16%. Notably, in GTSRB, 22 out of 43 classes benefit, with a maximum per-class improvement of

Table 1: Comparative evaluation of average accuracy of our proposed method (Ours) against baseline approaches across multiple datasets. The results report mean accuracy scores over three independent runs, with the best-performing values highlighted in **bold**. Entries with a gain of more than 1.5% over full-data training are highlighted in orange, while those exceeding 3% are shown in blue.

| Dataset | Original | Random | IF | TracIN | EWC | Trak | Ours |
|---|---|---|---|---|---|---|---|
| WaterBirds | 0.638 | 0.606 | 0.603 | 0.652 | 0.650 | 0.656 | **0.681** |
| AWA2-A | 0.644 | 0.622 | 0.644 | 0.652 | 0.642 | 0.638 | **0.662** |
| CELEBA | 0.895 | 0.893 | 0.890 | 0.893 | 0.890 | 0.898 | **0.906** |
| GTSRB | 0.969 | 0.966 | 0.973 | 0.971 | 0.975 | 0.971 | **0.980** |
| AWA2-B | 0.644 | 0.622 | 0.644 | 0.652 | 0.642 | 0.638 | **0.657** |
| CIFAR-10 | 0.774 | 0.787 | 0.798 | 0.784 | 0.789 | 0.793 | **0.801** |
| ImageNet-100 | **0.440** | 0.436 | 0.429 | 0.423 | 0.423 | 0.435 | 0.438 |

Table 2: Class-level accuracy improvement(Imp) after data removal across datasets. The table shows the maximum improvement in any class, the number of improved classes, and the mean improvement across them.

| Dataset | Max Imp | # Imp Classes | Mean Imp |
|---|---|---|---|
| Awa2-A | 16.27% | 6 / 10 | 11.12% |
| Awa2-B | 29.16% | 4 / 10 | 17.98% |
| CIFAR-10 | 10.39% | 7 / 10 | 5.59% |
| GTSRB | 50.00% | 22 / 43 | 5.69% |
| ImageNet-100 | 36.00 % | 51 / 100 | 10.15% |

50%. The improvement in average accuracy highlights that the improvement in underperforming classes is attained without substantially degrading the performance of other classes.

Table 3 reports gains in the worst-performing class for each dataset. In Awa2-A and Awa2-B, worst-class accuracy more than doubles, while in GTSRB, it improves by 20%. These results demonstrate that our method enhances class-level performance with minimal negative impact on other classes.

Table 3: Worst-class accuracy before and after retraining. The table shows the original worst-class accuracy and the corresponding value after retraining with the new dataset.

| Dataset | Original Worst-Class Accuracy | Retrained Worst-Class Accuracy |
|---|---|---|
| Awa2-A | 0.040 | 0.103 |
| Awa2-B | 0.040 | 0.103 |
| CIFAR-10 | 0.589 | 0.575 |
| GTSRB | 0.500 | 0.700 |
| ImageNet-100 | 0.100 | 0.100 |

## 4.4 Performance across Different Spurious Attributes

To further investigate the impact of spurious features on both worst-group and average performance, we follow the setup of Eyuboglu et al. (2022) and select a subset of the CELEBA dataset where the target attribute is strongly correlated with a spurious feature. We compare the average and worst-group performance achieved by our method against other baselines in Table 4 and Table 5. Additionally, considering the benefit of random data deletion in biased dataset (Chaudhuri et al., 2023) we introduce a new baseline, Maj.-Rand, where the subset of data is randomly deleted from the majority group. As shown in the results, our method outperforms other baselines in average accuracy in 7 and worst-group accuracy in 8 out of 10 settings, respectively. Notably, we observe a gain of over 4% in average accuracy for the target attribute attractive, compared to training on the original dataset. Similarly, worst-group accuracy improves by over 15% for attractive, receding hairline, and arched eyebrows, and by more than 5% for big nose, goatee, and male.

Table 4: Comparison of best average accuracy across different data attribution methods for different spurious attributes. The table reports the mean accuracy across three independent runs. Entries with a gain of more than 1.5% over full-data training are highlighted in orange, while those exceeding 3% are shown in blue.

| Target | Spurious Attribute | Original | Maj.-Rand | Random | IF | EWC | TracIN | Trak | Ours |
|---|---|---|---|---|---|---|---|---|---|
| arched eyebrows | receding hairline | 0.713 | **0.740** | 0.739 | 0.716 | 0.724 | 0.730 | 0.722 | 0.736 |
| attractive | mouth slightly open | 0.628 | 0.627 | 0.668 | 0.640 | 0.633 | 0.631 | 0.658 | **0.673** |
| big nose | male | 0.771 | 0.770 | 0.770 | 0.764 | 0.751 | 0.745 | 0.756 | **0.780** |
| goatee | bushy eyebrows | 0.946 | 0.931 | 0.947 | 0.938 | 0.951 | **0.953** | 0.949 | **0.953** |
| mouth slightly open | smiling | 0.869 | 0.871 | **0.877** | **0.877** | 0.860 | 0.876 | 0.867 | **0.877** |
| mouth slightly open | wearing lipstick | 0.820 | 0.804 | 0.801 | 0.828 | 0.834 | 0.816 | 0.801 | **0.839** |
| narrow eyes | eyeglasses | 0.840 | 0.858 | 0.862 | 0.856 | 0.858 | 0.860 | 0.855 | **0.862** |
| pointy nose | mouth slightly open | 0.690 | **0.714** | 0.676 | 0.689 | 0.695 | 0.709 | 0.694 | 0.698 |
| receding hairline | rosy cheeks | 0.921 | 0.909 | 0.920 | 0.921 | 0.920 | 0.916 | 0.911 | **0.930** |
| male | pointy nose | 0.919 | **0.931** | 0.907 | 0.909 | 0.911 | 0.906 | 0.915 | 0.921 |

Table 5: Comparison of best worst-group accuracy across different data attribution methods for different spurious attributes. The table reports the mean accuracy across three independent runs. Entries with a gain of more than 5% over full-data training are highlighted in green, while those exceeding 15% are shown in violet.

| Target | Spurious Attribute | Original | Maj.-Rand | Random | IF | EWC | TracIN | Trak | Ours |
|---|---|---|---|---|---|---|---|---|---|
| arched eyebrows | receding hairline | 0.187 | 0.314 | 0.113 | 0.247 | 0.262 | 0.196 | 0.099 | **0.354** |
| attractive | mouth slightly open | 0.213 | 0.242 | 0.347 | 0.266 | 0.241 | 0.205 | 0.392 | **0.407** |
| big nose | male | 0.131 | 0.076 | 0.096 | 0.143 | 0.092 | 0.113 | 0.172 | **0.221** |
| goatee | bushy eyebrows | 0.432 | 0.493 | 0.287 | 0.437 | 0.439 | 0.387 | 0.278 | **0.548** |
| mouth slightly open | smiling | 0.524 | 0.415 | **0.552** | 0.418 | 0.441 | 0.487 | 0.433 | 0.489 |
| mouth slightly open | wearing lipstick | 0.555 | 0.471 | 0.557 | 0.598 | 0.594 | 0.549 | 0.486 | **0.612** |
| narrow eyes | eyeglasses | **0.208** | 0.052 | 0.119 | 0.000 | 0.092 | 0.128 | 0.024 | 0.151 |
| pointy nose | mouth slightly open | 0.045 | 0.044 | 0.046 | 0.034 | 0.028 | 0.021 | 0.040 | **0.084** |
| receding hairline | rosy cheeks | 0.121 | 0.228 | 0.131 | 0.179 | 0.241 | 0.254 | 0.201 | **0.296** |
| male | pointy nose | 0.840 | 0.882 | 0.824 | 0.833 | 0.861 | 0.870 | 0.875 | **0.903** |

Table 6: Comparison of best average and best worst-group accuracy between metadata-driven and VLM-guided textual description.

| Target | Spurious Attribute | Meta Data | | VLM | |
|---|---|---|---|---|---|
| | | Avg. Acc. | WG Acc. | Avg. Acc. | WG Acc. |
| bangs | black hair | **0.922** | **0.649** | 0.916 | 0.624 |
| big nose | wearing necklace | **0.787** | **0.347** | 0.776 | 0.236 |
| heavy makeup | straight hair | 0.826 | **0.716** | **0.835** | **0.716** |
| wearing earrings | bags under eyes | **0.798** | **0.281** | 0.791 | 0.214 |

## 4.5 Ablation between Meta Data and VLM-based Description

Table 6 compares the performance of our method when using metadata-based versus VLM-generated textual descriptions of the spurious features. While both strategies yield comparable results in terms of average accuracy, the metadata-driven variant consistently achieves higher worst-group accuracy. This indicates that more precise annotations of underlying biases can facilitate the targeted removal of detrimental samples. Nevertheless, even in the absence of such annotations, VLM-based descriptions deliver comparable performance across both average and worst-group accuracies. These results suggest that, despite the lack of explicit metadata, VLM-generated descriptions do not degrade model performance or introduce additional bias.

Table 7: Comparison of best average and best worst-group accuracy between our method and D3M across different spurious attributes.

| Target | Spurious Attribute | Ours | | D3M | |
|---|---|---|---|---|---|
| | | Average Accuracy | Worst Group Accuracy | Average Accuracy | Worst Group Accuracy |
| bangs | black hair | **0.922** | **0.649** | 0.920 | 0.627 |
| big nose | wearing necklace | **0.787** | **0.347** | 0.747 | 0.173 |
| heavy makeup | straight hair | **0.826** | **0.716** | 0.821 | 0.654 |
| wearing earrings | bags under eyes | **0.798** | **0.281** | 0.787 | 0.068 |

## 4.6 Comparison with Group Annotation-based Subset Selection

Table 7 presents a comparative evaluation between our method, which relies on the textual description of bias, against a technique that can use group annotation of spurious features in the validation dataset. To compare with such a method, we define group structure based on different values of Spurious Attribute and Target, and then use the method proposed by Jain et al. (2024) (D3M) for subset selection. As per the result, on average, our method consistently outperforms D3M across both the best average and worst-group accuracy with a gain of 1.5% in best average accuracy and 11.8% in best worst group accuracy without using the explicit group annotation. This highlights the efficiency of the soft comparison scheme of clip features in handling partially visible features and the proposed optimization scheme compared to hard thresholding used in group annotation.

## 4.7 Ablation of Different Components

The ablation study in Table 8 highlights the contribution of key components i.e, data Attribution and CLIP, to the overall performance of our method. For the given experiment, we have used cosine similarity with CLIP (Only CLIP) representation to remove samples that align with the description of the underlying bias. When used independently, both components provide noticeable improvements over the full training baseline, particularly in average accuracy. However, they exhibit limitations in worst-group accuracy when applied in isolation. Notably, combining both Attribution and CLIP in our method yields the highest performance across nearly all settings, especially in worst-group accuracy, demonstrating the complementary strengths of these components in addressing spurious correlations.

Table 8: Comparative evaluation of the proposed method (Ours) with the full training baseline (Original), Only Attribution, and Only CLIP, reporting the best average and best worst group accuracy (mean) across three runs.

| Target Attribute | Spurious Attribute | Average Accuracy | | | | Worst Group Accuracy | | | |
|---|---|---|---|---|---|---|---|---|---|
| | | Original | Only Attribution | Only CLIP | Ours | Original | Only Attribution | Only CLIP | Ours |
| Bangs | Black Hair | 0.920 | 0.921 | 0.922 | **0.923** | 0.523 | 0.571 | 0.548 | **0.649** |
| Big Nose | Wearing Necklace | 0.765 | **0.787** | 0.777 | **0.787** | 0.127 | 0.080 | 0.110 | **0.347** |
| Heavy Makeup | Straight Hair | 0.805 | 0.800 | 0.813 | **0.826** | 0.651 | 0.686 | **0.739** | 0.716 |
| Wearing Earrings | Bags Under Eyes | 0.791 | 0.792 | 0.791 | **0.798** | 0.040 | 0.017 | 0.009 | **0.281** |

Table 9: Mean and standard deviation of detrimental attribution ($|\mathcal{A}|$) scores for different attributes, along with statistical significance from a two-sample t-test against *Smiling*.

| Attribute | Mean | Std | p-value (vs Smiling) | Significance |
|---|---|---|---|---|
| Smiling (spurious) | 0.539 | 0.056 | – | – |
| Moustache | 0.545 | 0.044 | 0.1008 | Not significant |
| Beard | 0.544 | 0.036 | 0.00039 | **Significant** |

### 4.8 Empirical Validation of Theoretical Formulation

To validate our theoretical claim, we used the codebase provided by Eyuboglu et al. (2022) to sample a 10k subset from CELEBA, where the attributes Male and Smiling are highly correlated. We then computed Trak scores for the training dataset using a ResNet-18 classifier trained to predict the Male label. In this setting, due to the strong correlation between Male and Smiling (Eyuboglu et al., 2022; Chen et al., 2023), smiling may act as a spurious feature. Since the task is to distinguish males from females, we consider features like Beard and Moustache to be more causally relevant, and thus expect that samples with these features to have lower $\mathcal{A}$ scores compared to those with Smiling.

However, statistical analysis of the detrimental attribution($\mathcal{A}$) scores using T-test for the training samples reveals that Smiling has lower scores for samples compared to samples with Beard and Moustache (Table 9). The difference is statistically significant for Beard ($p < 0.001$). This supports Proposition 1, demonstrating that such effects can arise in practical scenarios.

### 4.9 Comparison with Robustness based Methods

Table 10 compares the average and worst-group accuracy of our method against various robustness-based approaches on the Waterbirds dataset with ImageNet-based initialization(see Appendix I.3). Methods are grouped based on whether they require group annotations for the entire training dataset and whether they support textual descriptions of bias. As per the results, our method achieves a competitive average accuracy (0.855) and strong worst-group accuracy (0.756) without relying on group annotations for the training dataset, while uniquely supporting textual description of bias. Compared to other methods, our method improves worst-group accuracy by 27.9% over ERM, 12% over JTT, and 1.6% over D3M. Group annotation-based methods like gDRO and RWG perform best on worst-group accuracy, but at the cost of requiring explicit group labels for the entire training dataset. Although our method uses annotations for the validation dataset, either available through metadata or generated with the help of LLM or VLM; the identification of detrimental training samples relies only on the textual description of bias (inferred from the annotation), not on raw per-sample labels. This design can enable practical human-in-the-loop scenarios, where a domain expert can directly provide a textual description of the bias without relying on the annotated dataset. Additional challenges associated with robustness-based baselines in specific applications are discussed in Section 1, Section 2.2 and Appendix J.

Table 10: Comparison of Average Accuracy and Worst group accuracy achieved by our method in comparison with other robustness-based methods on Waterbirds.

| Method | Group Annotation (Train) | Supports Textual Bias Description | Average Accuracy | Worst Group Accuracy |
|--------|:---:|:---:|:---:|:---:|
| ERM | ✗ | ✗ | 0.819 | 0.477 |
| D3M | ✗ | ✗ | **0.903** | 0.740 |
| JTT | ✗ | ✗ | 0.852 | 0.636 |
| Ours | ✗ | ✓ | 0.855 | **0.756** |
| RWG | ✓ | ✗ | 0.864 | 0.822 |
| SUBG | ✓ | ✗ | 0.833 | 0.814 |
| gDRO | ✓ | ✗ | **0.886** | **0.836** |

## 5 Conclusion

In this work, we propose a data deletion framework to mitigate the impact of spurious biases in the training dataset and enhance model performance. Our method employs metric learning techniques to target and remove training samples that are semantically aligned with the textual description of identified biases and whose removal, based on attribution scores, does not adversely affect model performance. To the best of our knowledge, this is the first approach to use text-guided data attribution scores to mitigate simplicity bias in models. However, its effectiveness depends on the quality of the textual descriptions used to capture spurious biases, and the current framework is limited to image datasets. In future work, we aim to incorporate a human-in-the-loop framework to better mitigate complex biases and to extend it to NLP tasks.

## Acknowledgment

This research/project is supported (in part for setting up the GPU compute) by the Indian Institute of Science. Kumar Shubham is supported by the Kotak IISc AI-ML Centre (KIAC) fellowship.

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

## A Notations



Table 11: Notation table for key equation in main draft and proof



| Symbol | Description |
|---|---|
| **General Definitions** | |
| $h(x)$ | Model prediction for input $x$ |
| $y$ | True label corresponding to input $x$ |
| $\mathbf{1}(h(x) = y)$ | Indicator function: 1 if prediction is correct, else 0 |
| $\mathcal{D}_{\mathrm{val}}$ | Validation dataset |
| $\mathcal{D}_{(a^v, b_j^v)}$ | Subset of validation data with attribute–value pair $(a^v, b_j^v)$ |
| $\mathcal{D}_{\mathrm{train}}$ | Training dataset: $\mathcal{D}_{\mathrm{train}} = \{z_1, z_2, \dots, z_n\}$ where $z_i = (x_i, y_i)$ |
| $\mathcal{X}, \mathcal{Y}$ | Feature set $\mathcal{X} = \{x_1, x_2, \dots, x_n\}$ and label set $\mathcal{Y} = \{y_1, y_2, \dots, y_n\}$ |
| $z_k, z_i$ | Training samples from $\mathcal{D}_{\mathrm{train}}$ |
| $v_j$ | Validation sample |
| $\hat{y}$ | Output of the final logit layer of a neural network |
| $\theta$ | Vectorized parameters of the neural network, $\theta \in \mathbb{R}^p$ |
| $e_i$ | Standard unit vectors |
| **Neural Tangent Kernel (NTK) Specific** | |
| $\mathcal{G}(\mathcal{X}, \theta)$ | Neural Tangent Random Feature (NTRF) matrix: $\mathcal{G} = \frac{\partial \hat{y}(\mathcal{X}; \theta)}{\partial \theta}$ |
| $\mathcal{G}_0$ | NTRF matrix at initialization: $\mathcal{G}_0 = \mathcal{G}(\mathcal{X}, \theta_0)$ |
| **SVD Decomposition and Gradient Starvation** | |
| $U, S, V$ | Singular Value Decomposition (SVD) components: $Y\mathcal{G}_0 = USV^\top$ |
| $u^i, v_k$ | Singular vectors from $U$ and $V$ corresponding to features |
| $s_i$ | Singular value representing the strength of the $i^{th}$ feature |
| $\Gamma$ | Response of the network to features: $\Gamma = U^\top Y \hat{y} = SV^T \theta$ |
| $\Gamma_i$ | Response of the $i^{th}$ feature |
| **Attribution and Trak Scoring** | |
| $\Phi_n$ | Stacked gradient features of all training points for model $n$ |
| $\boldsymbol{f}(v; \theta)$ | Model output used for attribution (e.g., logit or loss) for input $v$ under parameters $\theta$ |
| $\boldsymbol{\phi}_n(\cdot)$ | Projected gradient feature under model $n$ |
| $\boldsymbol{\alpha}(v_j, z_i)$ | Attribution score: impact of removing $z_i$ on prediction for $v_j$ |
| $\mathcal{A}(z_i)$ | Detrimental attribution score for training sample $z_i$ |
| $p_n^{z_i}$ | Predicted probability for $z_i$ under model $n$ |
| $\mathcal{P}$ | Random projection matrix with entries drawn from $\mathcal{N}(0, 1)$ |
| **Optimization and others** | |
| $\mathcal{C}_{\mathrm{text}}, \mathcal{C}_{\mathrm{image}}^i$ | Text and image embeddings respectively |
| $\mathcal{T}$ | Trade-off hyperparameter: $\mathcal{T} = \beta \times |\mathcal{D}_{\mathrm{train}}|$ (control tradeoff between data attribution and semantic coherence) |
| $\beta$ | Hyperparameter associated with $\mathcal{T}$ |
| $k_i$ | Selection weight for sample $i$ |
| $d(k)$ | Penalty term enforcing deletion constraint |
| $\mathcal{L}$ | Final optimization objective including penalty |
| C | Hyperparameter associated with soft penalty Equation 9 |
| $M = LL^\top$ | Metric matrix constructed from $L$ |
| $L \in \mathbb{R}^{D \times t}$ | Learnable matrix under optimization defined by Equation 7 |
| $\tau$ | Accuracy threshold to detect spurious bias |
| $\theta^*(\cdot)$ | Final model parameters trained on the specified dataset |

## B Details on Trak

$$\boldsymbol{\alpha}(v_j, z_i) = \frac{1}{N} \sum_{n=1}^{N} \left( \boldsymbol{\phi}_n(v_j)^\top (\Phi_n^\top \Phi_n)^{-1} \boldsymbol{\phi}_n(z_i) \right) \times \frac{1}{N} \sum_{n=1}^{N} \left( 1 - p_n^{z_i} \right)$$

$$\text{where, } p_n^{z_i} = (1 + \exp(-y_i \boldsymbol{f}(x_i; \theta_n^*)))^{-1}, \boldsymbol{\phi}_n(v_j) = \mathcal{P}^\top \nabla_\theta \boldsymbol{f}(v_j; \theta_n^*),$$

$$\boldsymbol{\phi}_n(z_i) = \mathcal{P}^\top \nabla_\theta \boldsymbol{f}(z_i; \theta_n^*), \ \Phi_n = \left[ \boldsymbol{\phi}_n(z_1)^\top; \dots; \boldsymbol{\phi}_n(z_{|\mathcal{D}_{\mathrm{train}}|})^\top \right]$$

$$\Phi_n \in \mathbb{R}^{m \times k}, \mathcal{P} \sim \mathcal{N}(0, 1)^{p \times k}, \quad k \ll p. \tag{8}$$

Equation 8, illustrates the calculation of the trak score. Scores consist of an average of the data attribution score calculated over multiple checkpoints (N). The terms $\phi_n(v_j)$ and $\phi_n(z_i)$ denote the projected gradients of the validation sample $v_j$ and the training sample $z_i$ for the $n^{th}$ set of parameters and projection matrix $\mathcal{P}$. This projection matrix reduces the dimension of the gradient $\nabla_\theta \boldsymbol{f}(z; \theta_n^*) \in \mathbb{R}^p$ to a lower-dimensional space $\mathbb{R}^k$, where $k \ll p$, while approximately preserving the inner product, as per the classical Johnson-Lindenstrauss theorem (Johnson, 1984).

## C  Algorithm

---

**Algorithm 1** Proposed Method

---

**Require:** Training Dataset ($\mathcal{D}_{\text{train}}$), Validation Dataset ($\mathcal{D}_{\text{valid}}$), Number of Checkpoints ($M$), Rank for Metric Learning ($t$), Min Weight Fraction ($\beta$), Cutoff for Subset Selection ($\gamma$), CLIP Embedding Model ($\mathcal{C}$), Epochs for Classifier Training ($\mathcal{E}$), Optimization Iterations for Metric Learning ($\mathcal{I}$).

1: ## **Classifier Training**
2: i=0
3: **for** epoch $\in [0 \ldots \mathcal{E}]$ **do**
4:     Train the classifier using $\mathcal{D}_{\text{train}}$.
5:     **if** epoch $\in [\mathcal{E}, \mathcal{E} - 2, \mathcal{E} - 4, \mathcal{E} - 6, \mathcal{E} - 8]$ **then**
6:         Save the checkpoint $\theta_i$.
7:         i+=1
8:     **end if**
9: **end for**
10: Save $N = [\theta_0, \theta_1, \theta_2, \theta_3, \theta_4]$ checkpoints for the calculation of attribution score as per Equation 4.
11: ## **Spurious Feature Identification**
12: Generate a list of possible attributes and corresponding values for $\mathcal{D}_{\text{valid}}$ using ChatGPT (Section 3.2).
13: **for** $i \in [1 \ldots |\mathcal{D}_{\text{val}}|]$ **do**
14:     Annotate attribute-value pairs for sample $v_i$ using a Llama-based VLM model (Section 3.2).
15: **end for**
16: ## **Calculating the Detrimental Attribution Score**
17: **for** $z_i \in \{z_1 \ldots z_n\}$ **do**
18:     Calculate the attribution score $\mathcal{A}(z_i)$ using the saved checkpoints (Equations 4 and 5).
19: **end for**
20: Compare the accuracy of each attribute-value pair using Equation 1. Flag an attribute-value pair as spurious if its accuracy exceeds the average dataset accuracy by a threshold $\tau$.
21: Generate a textual representation of flagged attribute-value pairs under the context of the dataset (Appendix I.5).
22: Create a CLIP embedding of the textual representation ($\mathcal{C}_{\text{text}}$).
23: ## **Metric Learning**
24: **for** $i \in [1 \ldots |\mathcal{D}_{\text{train}}|]$ **do**
25:     Calculate the CLIP image embedding ($\mathcal{C}_{\text{image}}^i$) for each sample $z_i$ in $\mathcal{D}_{\text{train}}$.
26: **end for**
27: **for** $i \in [0 \ldots \mathcal{I}]$ **do**
28:     Optimize the loss $\mathcal{L}$ using $\mathcal{C}_{\text{text}}$, $\mathcal{C}_{\text{image}}$, and $\mathcal{A}(z)$ as per Equation 10 to generate the metric $\mathbf{k}$ using the hyperparameter t, $\beta$.
29: **end for**
30: Use the score $\mathbf{k}$, $\gamma$ to identify $\mathcal{S}^{\text{deter}}$ and retrain the model on $\mathcal{D}_{\text{train}} \setminus \mathcal{S}^{\text{deter}}$.

---

## D  Soft Penalty for Optimization

For efficient optimisation of the constrained objective presented in Equation 7, we have replaced the hard constraint with a soft constraint ( d(k) ) as per d'Eon et al. (2022).

$$d(k) = C \cdot \max \left( \frac{\left( \sum_{i=1}^{|\mathcal{D}_{\text{train}}|} k_i - (\mathcal{T} + w) \right)^2}{w^2}, 0 \right), \tag{9}$$

This penalty term is quadratic and scaled by a shrinkable weight w, which is gradually reduced throughout the optimization process. The overall unconstrained optimization problem is defined in Equation 10 where C is a hyperparameter.

$$\mathcal{L} = \max_L \sum_{i=1}^{|\mathcal{D}_{\text{train}}|} \left( \frac{k_i}{\sum_j k_j} \right) \mathcal{A}(z_i) - d(k). \tag{10}$$

## E  Theoretical Formulation

For dataset $\mathcal{D}_{\text{train}} = \{z_1 \dots z_n\}$ where $z_i = (x_i, y_i)$, and $\boldsymbol{x}_i \in \mathbb{R}^d$, and corresponding labels $y_i \in \{-1, +1\}^n$. Let $\hat{y}$ denotes the output of the final logit layer of an L-layer neural network trained using binary cross-entropy, and $\theta \in \mathbb{R}^p$ represents a p-dimensional vectorized parameter of the neural network (Equation 4, Equation 8). let $\mathcal{X} = \{x_1 \dots x_n\}$ and $\mathcal{Y} = \{y_1 \dots y_n\}$ constitute the respective features and class labels.

In the Neural Tangent Kernel (NTK) framework (Jacot et al., 2018), the final output of a neural network can be approximated as a linear function of parameters, whose properties are governed by the Neural Tangent Random Feature (NTRF) matrix, defined as:

$$\mathcal{G}(\mathcal{X}, \theta) = \frac{\partial \hat{y}(\mathcal{X}; \theta)}{\partial \theta}, \quad \mathcal{G} \in \mathbb{R}^{n \times p}. \tag{11}$$

For wide-width neural networks, the NTRF matrix remains approximately constant during training (Pezeshki et al., 2021), allowing the output of the neural network to be approximated using the initial NTRF matrix, $\mathcal{G}_0 = \mathcal{G}(\mathcal{X}, \theta_0)$, as follows:

$$\hat{y}(\mathcal{X}, \theta) = \mathcal{G}_0 \theta. \tag{12}$$

The dominant features of the dataset can be estimated using the principal components of $\mathcal{G}_0 = \mathcal{G}(\mathcal{X}, \theta_0)$, which are equivalent to the principal components of the NTK gram matrix (Yang & Salman, 2019).

**Definition 2** (Features and gradient starvation (Pezeshki et al., 2021))**.** *Consider a support vector decomposition of $Y\mathcal{G}_0 = USV^\top$, where $Y = diag(y)$, the $i^{th}$ feature is represented by $(V^\top)_{(i,:)}$ or $(V)_{(:,i)}$ with its strength denoted as $s_i = (S)_{ii}$ and its weight across all training samples represented by $(U)_{(:,i)}$. The response of the neural network to the $i^{th}$ feature can be expressed as $\Gamma_i$ , where:*

$$\Gamma := U^\top Y \hat{y} = SV^T \theta.$$

*Due to the imbalance in the training dataset, for a given set of features and the optimal parameter $\theta^*$, the presence of the $i^{th}$ feature can influence the learning of the $j^{th}$ feature. This phenomenon, referred to as gradient starvation, arises in optimal parameters if:*

$$\frac{d\Gamma_j^*}{d(s_i^2)} < 0$$

Definition 2 suggests that as the strength of the $i^{th}$ feature $(s_i^2)$ increases, the learning of the $j^{th}$ feature gets impacted. This implies that stronger features can dominate the learning process, leading to a reduced contribution of other informative features in the model's predictions.

**Definition 3** (Feature contribution and strength for a given sample). *Following Definition 2 and the formulation of Pezeshki et al. (2021), consider a training instance $z_k$ and a feature $f_j$. We define the (instance-specific) contribution of feature $f_j$ to the sample $z_k$ as $u_j^k$ ($U_{k,j}$) and the strength of the feature in the neural network as $s_{j,j}$.*

**Theorem 1** (Gradient Starvation Regime (Pezeshki et al., 2021)). *For a neural network in the linear regime and trained using binary cross entropy loss with feature coupling between two features $f_1$ and $f_2$ as defined in Pezeshki et al. (2021) and with $s_1^2 > s_2^2$, we have,*

$$\frac{d\Gamma_2^*}{d(s_1^2)} < 0,$$

Now, under the given setting, we will try to understand the influence of gradient starvation on the performance of the NTK-based data attribution methods :

**Proposition 2** (Illustration for the Under Valuation of Attribution Scores). *Consider a neural network in the neural tangent kernel (NTK) regime, trained using binary cross-entropy loss and contain two important features $f_1$ and $f_2$. Suppose that due to learning dynamics, $f_1$ becomes dominant and causes gradient starvation of $f_2$ as per Pezeshki et al. (2021). Let $z_i$ and $z_j$ be two training samples for which $f_1$ and $f_2$ are the most informative features, respectively, and assume that the they contain equal representations of these features (equivalently, $u_1^i = u_2^j$ as per Definition 3). Under these conditions, the detrimental attribution score for $z_i$ can be systematically undervalued relative to $z_j$. Formally:*

$$\big|\mathcal{A}(z_i)\big| < \big|\mathcal{A}(z_j)\big|$$

*Proof.* For a sigmoid-based activation, the output probability for feature set $\big(\mathcal{X}\big)$ is given by:

$$
\begin{aligned}
p(\mathcal{X};\theta) &= \frac{1}{1 + \exp\big(-\hat{y}(\mathcal{X};\theta)\big)}, \\
p(\mathcal{X};\theta) \cdot \big(1 + \exp(-\hat{y}(\mathcal{X};\theta))\big) &= 1, \\
p(\mathcal{X};\theta) \cdot \exp\big(-\hat{y}(x;\theta)\big) &= 1 - p(\mathcal{X};\theta), \\
\hat{y}(\mathcal{X};\theta) &= \log\left(\frac{p(\mathcal{X};\theta)}{1 - p(\mathcal{X};\theta)}\right).
\end{aligned}
\tag{13}
$$

Hence, the utility function ($\boldsymbol{f}$) used in Trak for data attribution (Equation 3) is equivalent to the logit of a binary cross entropy ($\hat{y}$).

The gradient of the logit under the NTK framework gives,

$$\frac{\partial \hat{y}(x;\theta)}{\partial \theta} = \frac{\partial \mathcal{G}_0 \cdot \theta}{\partial \theta} = \mathcal{G}_0 \tag{14}$$

As per Equation 4 and Equation 8, $\Phi_m = \mathcal{G}_0 \cdot \mathcal{P}$. Now, considering that the projection matrix (Johnson, 1984) preserves the inner product of the actual gradient vector. We will simplify our argument and calculate the value for the unprojected gradients (Park et al., 2023) $\big(\mathcal{P} = I_d\big)$. Furthermore, under the NTK regime, where the optimal parameters are similar (Jacot et al., 2018), we calculate the attribution score for a single checkpoint (M=1). For ease of derivation, we will omit the subscript m i.e., $\Phi_1 = \Phi$ and $\phi_1 = \phi$, hence:

$$\Phi = \mathcal{G}_0 \tag{15}$$
$$\Phi^T \Phi = \mathcal{G}_0^T \mathcal{G}_0 \tag{16}$$

Now, as per the feature decomposition defined in Definition 2 :

$$Y\mathcal{G}_0 = USV^\top$$

$$(Y\mathcal{G}_0)^\top (Y\mathcal{G}_0) = \left(USV^\top\right)^\top \left(USV^\top\right)$$

$$\mathcal{G}_0^T Y^\top Y\mathcal{G}_0 = VS^2V^T$$

(17)

Since $Y = \operatorname{diag}\{y_1,\ldots,y_n\}$ and $y \in \{-1,1\}$, it follows that:

$$Y^T Y = I,$$

$$\mathcal{G}_0^T \mathcal{G}_0 = VS^2V^T,$$

$$\Phi^T \Phi = VS^2V^T.$$

(18)

The validation attribution score (Equation 5) is given by :

$$\mathcal{A}(z_i) = \sum_{v_j \in \mathcal{D}_{val}} -\alpha(v_j; z_i)$$

$$= \sum_{v_j \in \mathcal{D}_{val}} -\phi(v_j)^\top (\Phi^\top \Phi)^{-1} \phi(z_i)(1 - p^{z_i})$$

(19)

Substituting the value of $\Phi^T \Phi$:

$$\mathcal{A}(z_i) = \sum_{v_j \in \mathcal{D}_{val}} -\phi(v_j)^\top (VS^2V^T)^{-1} \phi(z_i)(1 - p^{z_i})$$

$$= \left(\sum_{v_j \in \mathcal{D}_{val}} -\phi(v_j)^\top\right)(V)^{-1^\top} S^{-2}(V)^{-1} \phi(z_i)(1 - p^{z_i})$$

$$= \left(\sum_{\boldsymbol{v}_j \in \mathcal{D}_{val}} -\phi(v_j)^\top\right) VS^{-2}V^\top \phi(z_i)(1 - p^{z_i})$$

$$= \left(\sum_{\boldsymbol{v}_j \in \mathcal{D}_{val}} -\nabla_\theta f(\boldsymbol{v}_j,\theta)^\top\right) VS^{-2}V^\top \nabla_\theta f(z_i,\theta)(1 - p^{z_i}) \text{ (since } \mathcal{P} = I \text{ and as per Equation 8 )}$$

$$= \left(\sum_{\boldsymbol{v}_j \in \mathcal{D}_{val}} -\nabla_\theta f(\boldsymbol{v}_j,\theta)^\top\right) VS^{-2}V^\top \nabla_\theta f(z_i,\theta)(1 - p^{z_i})$$

$$= \sum_k \frac{\left(\left(\sum_{\boldsymbol{v}_j \in \mathcal{D}_{val}} -\nabla_\theta f(\boldsymbol{v}_j,\theta)^\top\right)v_k\right)\left(v_k^\top \nabla_\theta f(z_i,\theta)(1 - p^{z_i})\right)}{s_{kk}^2}$$

(20)

where, $v_k$ is the $k^{th}$ column of V matrix and representing the $k^{th}$ feature as per Definition 2

now given the definition of the $\mathcal{G}_0$ and as per Equation 8, Equation 13 and Equation 15

$$\mathcal{G}_0 = [\nabla_\theta f(z_1,\theta)^\top;\ldots;\nabla_\theta f(z_n,\theta)^\top)]$$

$$Y\mathcal{G}_0 = USV^\top$$

(21)

For the $i^{th}$ training sample, this score can be further simplified by multiplying with the standard unit vector ($e_i$) on both sides:

$$e_i^\top Y \mathcal{G}_0 = e_i^\top U S V^\top$$
$$y_i \nabla_\theta f(z_i, \theta)^\top = u^i S V^\top$$

$$(22)$$

where $u^i$ is a row vector associated with matrix U,

multiplying both side with $y_i$ and $V$ we get ,

$$y_i \cdot y_i \nabla_\theta f(z_i, \theta)^\top V = y_i u^i S$$

as $y_i^2 = 1$ and further multiplying both side with $e_k$ we get

$$\nabla_\theta f(z_i, \theta)^\top V \cdot e_k = y_i u^i S \cdot e_k$$
$$\nabla_\theta f(z_i, \theta)^\top v_k = y_i u_k^i s_{kk}$$

$$(23)$$

substituting the value in Equation 20 gives :

$$|\mathcal{A}(z_i)| = \left| \sum_k \frac{\left( \sum_{\boldsymbol{v}_j \in \mathcal{D}_{val}} -\nabla_\theta f(\boldsymbol{v}_j, \theta)^\top \right) v_k y_i u_k^i \left(1 - p^{z_i}\right)}{s_{kk}} \right|$$

$$(24)$$

According to the given equation under similar probability measure, for any two data points $z_i$ and $z_j$ where the dominant features are $f_1$ and $f_2$ respectively, the contribution of these features, as per Definition 2, 3, is represented by $u_1^i$ and $u_2^j$. When both dominant features are equally represented, it follows that $u_1^i = u_2^j$ and $u_1^j < u_2^j$, $u_2^i < u_1^i$ . Furthermore, if $|s_{11}| > |s_{22}|$ then as per Theorem 1 $f_1$ induces gradient starvation of $f_2$ and results in lower detrimental attribution score i.e., $|\mathcal{A}(z_i)| < |\mathcal{A}(z_j)|$. $\qquad \square$

### E.1 Discussion on Real-World Implications of the Analysis

As shown in our analysis, the detrimental attribution scores for samples $z_i$ and $z_j$ dominated by features $f_1$ and $f_2$, respectively, are governed by the ratios

$$\frac{u_1^i}{s_{11}} \quad \text{and} \quad \frac{u_2^j}{s_{22}},$$

where $u_1^i$ and $u_2^j$ denote the contributions of the respective features, and $s_{11}$ and $s_{22}$ capture their effective strengths.

When $|s_{11}| > |s_{22}|$, *gradient starvation* occurs i.e., the feature $f_1$ dominates the training dynamics and suppresses the learning of feature $f_2$. For samples $z_i$ and $z_j$ with equally informative features (i.e., $u_1^i = u_2^j$), this leads to

$$\frac{u_1^i}{s_{11}} < \frac{u_2^j}{s_{22}},$$

resulting in lower attribution scores for samples $z_i$ dominated by the spurious feature $f_1$ (the feature inducing gradient starvation) compared to samples $z_j$ dominated by the causal feature $f_2$.

In realistic settings, samples often predominantly exhibit either causal or spurious features, for example, $u_2^i \approx 0$ for $z_i$ and $u_1^j \approx 0$ for $z_j$. Even in such cases if the causal feature contribution exceeds that of the spurious feature (i.e., $u_2^j > u_1^i$) the inequality

$$\frac{u_1^i}{s_{11}} < \frac{u_2^j}{s_{22}}$$

can still hold. As a result, samples dominated by *causal features* may be preferentially removed. Intuitively, gradient starvation causes the model to overestimates the relevance of spurious features and incorrectly treats samples containing causal information as detrimental.

## F   Responsible AI

This work addresses responsible AI concerns arising from simplicity biases and distributional imbalances in large-scale training datasets, which are often collected from heterogeneous and weakly curated sources. Such biases can cause models to rely on spurious features, resulting in unfair performance degradation across subpopulations and unreliable behavior under distribution shifts.

The current work presents a data-centric framework for identifying and mitigating spurious correlations directly at the data level. Our approach aims to improve robustness and reduce unintended discriminatory effects without altering model architectures or training protocols, which may be constrained by regulatory or safety requirements. Notably, while our framework employs VLMs solely for generating high-level textual descriptions of spurious features, their use can be entirely bypassed when suitable metadata or domain expert knowledge is available. Moreover, by controlling the trade off between attribution score and textual description of bias (controlled via the weighting parameter $\beta, \mathcal{T}$ in Equation 7) one can ensures that potential biases from auxiliary models like VLMs have minimal influence on the debiasing process.

Our theoretical and empirical results indicate that attribution scores can be affected by underlying spurious features, suggesting that it is an important consideration when using attribution-based methods. For all our experiment we have only used publicly available dataset. As such, this work contributes toward more transparent and robust machine learning systems, particularly in high-stakes settings where biases in performance can have a significant impact.

## G   Group-wise Accuracy Improvements

Table 12 presents the group-wise accuracy before and after removing $\mathcal{S}^{\mathrm{deter}}$ from the training dataset using our proposed method. Groups 1–4 report the model's performance when trained on the original dataset for different combinations of target and spurious features, whereas Groups 1*–4* show the corresponding results after updating the dataset ($\mathcal{D}_{\mathrm{train}} \setminus \mathcal{S}^{\mathrm{deter}}$). As shown in the results, our method yields notable improvements in several groups without significant degradation in others. This demonstrates that the proposed approach improves the performance of the worst-performing group while maintaining comparable accuracy across the remaining groups, thereby avoiding the introduction of new biases.

Table 12: Group-wise accuracy before and after removing spurious samples. The table reports the mean accuracy and standard deviation over 3 runs. Groups 1–4 represent the training with the original dataset, while Groups 1*–4* correspond to results after data pruning.

| Target Attr | Spurious-Attr | G1 | G2 | G3 | G4 | G1* | G2* | G3* | G4* |
|---|---|---|---|---|---|---|---|---|---|
| Bangs | Black Hair | 0.73 | 0.97 | 0.53 | 0.98 | 0.78 | 0.96 | 0.59 | 0.97 |
| Big Nose | Necklace | 0.13 | 0.94 | 0.32 | 0.89 | 0.22 | 0.94 | 0.37 | 0.89 |
| Heavy Makeup | Straight Hair | 0.684 | 0.80 | 0.81 | 0.82 | 0.72 | 0.83 | 0.80 | 0.80 |
| Earrings | Bags Under Eyes | 0.09 | 0.99 | 0.04 | 0.99 | 0.28 | 0.96 | 0.32 | 0.91 |

## H   Data Annotation

### H.1   Attribute Generation

We utilize ChatGPT to generate attributes for a specific dataset with the following prompt referenced from HiBug (Chen et al., 2024). The list of attribute-value pairs generated by ChatGPT is provided in Table 13.

*You are a helpful assistant to help user work on improving AI visual models. You need to discuss with your user for a description of the task that the model is working for. You need to decide if the description is complete and clear enough. The description should at least contains or infer the task object, task type, task scene. After understanding user's task description, you should generate related visual attributes that might*

*affect the model's performance. You should not ask me to provide visual attributes. (Note that this is only an example visual attributes according to the previous example, do not take any of its values as default value!): "Gender , Age , Hairstyle , Hair colour" If user is satisfied with the attributes, generate the attribute form with the header formatted as "//Attribute Form//" and end with "//END//". Attributes in the form should be splited by comma. Do not include the task object, task type, task scene. (Note that this is only an example visual attributes according to the previous example, do not take any of its values as default value!): //Attribute Form// Gender , Age , Hairstyle , Hair colour //END//*

Table 13: Details of the attribute value pair generated using ChatGPT.

| Dataset | Attributes | Choices |
|---|---|---|
| AWA2 | Size of the Animal | Small, Medium, Large, Very Large |
| | Fur or Skin Texture of Animals | Smooth, Rough, Furry, Scaly |
| | Color Pattern on Animal | Striped, Spotted, Solid Color, Mixed Colors |
| | Posture of Animal | Sitting, Standing, Flying, Running |
| | Visible Markings or Patterns | Scars, Spots, Unique Patterns |
| | Lighting Conditions | Bright, Dim, Natural, Artificial, Shadowy |
| | Background Complexity | Plain, Cluttered, Natural Habitat |
| | Presence of Humans | None, Nearby, Interacting |
| | Animal Activity State | Resting, Moving, Feeding, Playing |
| | Occlusions | Fully Visible, Partially Hidden |
| | Weather Conditions | Sunny, Cloudy, Rainy, Foggy, Snowy |
| | Seasonal Variations | Summer Coat, Winter Coat, Shedding Fur |
| CELEBA | Gender | Male, Female |
| | Age | Child, Teenager, Adult, Elderly |
| | Facial Expression | Neutral, Smiling, Frowning, Surprised |
| | Hairstyle | Short, Long, Bun, Braided |
| | Hair Color | Black, Brown, Blonde, Red |
| | Skin Tone | Light, Medium, Dark |
| | Facial Hair | Beard, Mustache, Clean-shaven |
| | Presence of Accessories | Glasses, Earrings, Necklace |
| | Lighting Conditions | Bright, Dim, Shadowed |
| | Makeup | Natural, Heavy, None |
| CIFAR-10 | Size | Large, Medium, Small |
| | Pose/Orientation | Side View, Top View, Angled |
| | Lighting | Daylight, Nighttime, Shadows |
| | Background Complexity | Plain, Crowded |
| | Object Occlusion | Partially Visible, Fully Visible |
| GTSRB | Shape of Sign | Round, Triangular, Rectangular |
| | Color of Sign | Red, Blue, Yellow, White |
| | Size of Sign | Small, Medium, Large |
| | Weather Conditions | Sunny, Rainy, Foggy, Overcast |
| | Lighting | Daylight, Nighttime, Shadows, Glare |
| WaterBirds | Surrounding Environment | Forest Floor, Beach, Lake, River, Ocean, Shoreline |
| | Background Elements | Trees, Bushes, Rocks, Water Bodies, Sand, Human-made Structures |
| | Lighting Conditions | Full Daylight, Shaded Areas, Low-light, Overcast |
| | Weather Conditions | Sunny, Cloudy, Rainy, Foggy, Windy |

## H.2 Attribute-Value Annotation

We employ Llama 3.2 (Dubey et al., 2024), a Vision-Language Model (VLM) with 11B parameters, to determine the most suitable value among a set of possible attributes and values for a given dataset. By iterating over a set of images in the validation set, the VLM generates metadata, which is subsequently utilized to identify the spurious features. Each image approximately takes 4-10 seconds on average to annotate, depending on the size of the image. The system prompt provided to Llama 3.2 is as follows:

*You are an expert in identifying visual attributes in a given image. You will be presented with an image along with attributes and a list of choices for each of the attributes. You will be asked to choose the most suited choice for each of the attributes present in the image. Only choose one choice among all given choices for a particular attribute. Ensure that the choice is a string. Reproduce the attribute and the choice as it is. Preserve the case and the spelling. Respond with only a valid JSON object with the attributes as the keys and the chosen choices as the values, and no other extra fluff. Use double inverted commas.*

# I  Training Procedure

## I.1  Model Configuration and Metrics

We maintained consistent hyperparameter settings across all baselines, with the only variation being the subset of training data selected by each method. The validation set was used to identify underlying spurious biases, as outlined in Section 3.2. For baseline comparisons, we utilized publicly available implementations. In cases where the code was not open-sourced or experiments were not conducted on the specific datasets, we implemented the methods and used the respective datasets for evaluation. For TracIN, we employed the fast implementation available in the Captum library (Kokhlikyan et al., 2020).

Since many real-world datasets lack well-defined group structures (Sagawa et al., 2020), which are typically needed for evaluating worst-group accuracy, we compare our method and baselines primarily on average accuracy. Additionally, to understand the influence of deleting data samples in mitigating spurious features, we follow the experiment setup defined by (Chaudhuri et al., 2023; Idrissi et al., 2022; Sagawa et al., 2020) and analyze the worst-case performance improvement. We used the methodology proposed in (Eyuboglu et al., 2022) to create a subset of CELEBA with specific simplicity biases.

## I.2  Model Training and Datasets

All experiments reported in Table 1 were conducted using the ResNet-18 architecture. The models were trained from scratch with random initialization. For the WaterBirds dataset, the classifier was trained for 15 epochs using stochastic gradient descent with a momentum value of 0.9 and a learning rate of 0.001. For all other datasets, we used the Adam optimizer with a learning rate of 0.001.

For AWA2-A, AWA2-B, CELEBA, datasets, models were trained for 15 epochs, while for GTSRB and CIFAR-10, models were trained for 5 epochs. We have used the same 10 classes as mentioned in Boecking et al. (2022) for all experiments related to AWA2. For CELEBA, we used a subset of 10,000 examples from the original dataset, with the target label being hair color (blond) and the spurious feature being gender (male). Additionally, we induced a spurious correlation of 0.4 between the target and spurious features to mimic real-world biases. For experiments related to ImageNet-100, we have considered the subset of the ImageNet dataset with 100 classes as per Tian et al. (2020) and trained the model for 10 epochs with the Adam optimizer. We have further considered the attributes related to texture and shape for common classes available for the ImageNet dataset (Russakovsky & Fei-Fei, 2010). The cutoff value to mark an attribute-value pair as spurious ($\tau$) was decided based on the size of the corresponding pair in the validation dataset, and the pair generating the largest difference with respect to the original dataset was picked for analysis.

To ensure a fair comparison for subset selection, we maintained uniformity in the training process across both the original model training and the retraining process after data deletion.

The experiments reported in Table 4, Table 5 were conducted using the ResNet-18 model, trained for 10 epochs with the Adam optimizer and a learning rate of 0.001. The dataset was created by randomly sampling the correlation factor within the range [0,1] and varying the training data size across [5000, 3000, 7000, 10000]. The correlation attribute and target attribute were selected from the metadata provided in the CELEBA dataset (Eyuboglu et al., 2022). Experiments on the following target–correlated attribute pairs—(arched eyebrows, receding hairline), (attractive, mouth slightly open), (big nose, male), (goatee, bushy eyebrows), (mouth slightly open, smiling), (mouth slightly open, wearing lipstick), (narrow eyes, eyeglasses), (pointy nose, mouth slightly open), (receding hairline, rosy cheeks), and (male, pointy nose) are conducted with

varying training dataset sizes of 3000, 5000, 5000, 5000, 5000, 5000, 7000, 7000, 7000, and 5000 samples respectively, and corresponding spurious correlation strengths of 0.2, 0.8, 0.4, 0.4, 0.8, 0.9, 0.2, 0.6, 0.6, and 0.6 respectively. Further experiments on the target attributes Bangs, Big Nose, Heavy Makeup, and Wearing Earrings, were conducted with correlation factors of 0.6, 0.2, 0.4, and 0.2, and with training sample sizes of 10000, 5000, 3000, and 5000, respectively. Results for these experiments are provided in Table 8.

### I.3    Comparison with Other Optimization and Data-Centric Methods

In general, ImageNet initialization (Pham et al., 2021) plays a crucial role in achieving strong worst-group accuracy. However, most of our experiments are conducted without ImageNet pretraining to better reflect practical deployment scenarios, particularly those where spurious correlations can significantly degrade model performance (Pham et al., 2021). For a fair comparison with optimization-based methods such as gDRO (Sagawa et al., 2020) and JTT (Liu et al., 2021), we additionally evaluate our method on the Waterbirds dataset using a ResNet-18 model pretrained on ImageNet, along with LLM-generated attribute–value annotations. Results averaged over three independent runs are reported in Table 10. We also include comparisons with data deletion methods like D3M (Jain et al., 2024) and group-balancing approaches such as SUBG and RWG (Idrissi et al., 2022).

### I.4    Data Attribution and Subset Size

For the experiments reported in Table 1, approximately 3% of the data was removed from the training dataset. We fix the data removal budget across all baselines, as it is a design choice best left to domain experts. A smaller removal percentage prevents overpruning of the dataset ( training sample for group land bird on water is around 56 out of 4795 (Idrissi et al., 2022) ) and highlights the precision of attribution methods by focusing on the most harmful samples. In contrast, larger removals can obscure differences between methods due to overlapping sample selections. For experiments related to spurious correlation in celeba, considering the stochasticity of the training sample, we have fixed the budget size to 100 samples. Further ablation on subset size is provided in Appendix P. We ensured uniformity in the data deletion process by basing it on the validation attribution score $\mathcal{A}$, calculated according to the respective definition of data attribution $\alpha$ in each baseline method, using their default hyperparameters.

For our proposed method, we performed hyperparameter tuning by selecting the rank parameter ($t$) from [50, 40, 10, 100] and the minimum weight ($\beta$) from [0.6, 0.7, 0.8, 0.9, 0.95]. The weight barrier ($C$) was chosen from [5, 10]. The optimization for Equation 10 was performed for 5000 iterations using the Adam optimizer with a learning rate of 0.0001. The value of $\gamma$ is decided based on the fraction of the dataset that is removed from the training dataset. For experiments reported in Table 4 and Table 5, hyperparameter tuning was performed over the same range as in previous experiments, optimizing for both best average performance and best worst-group accuracy separately.

### I.5    Textual Description

For different datasets, we used distinct textual representations of the underlying bias. The choice of textual descriptions in our experiments depends not only on the attribute-value pairs but also on the dataset itself. For instance, datasets like AWA2-A contain only label-specific information, such as color and habitat type, without an explicit attribute-value format. Therefore, a suitable textual representation for this dataset could be *"It is a (*1) animal."* Here, (*1) represents the feature identified as a potential biased candidate. Similarly, for GTSRB, incorporating dataset context improves model performance, and a possible template could be *"(*1) of the sign is (2)."* where (*1) and (*2) are replaced by the corresponding attribute and value pair.

For datasets such as WaterBirds, AWA2-A, AWA2-B, CELEBA, GTSRB, CIFAR-10, and ImageNet-100 the textual descriptions used in the experiments related to Table 1 are provided in Table 14:

Table 14: Textual descriptions of spurious feature for different datasets

| Attribute Description | Dataset |
|---|---|
| *Surrounding environment in image is forest floor* | WaterBirds |
| *It is a domestic animal* | AWA2-A |
| *Size of the animal is very large* | AWA2-B |
| *Image of a male with blond hair* | CELEBA |
| *Shape of the sign is round* | GTSRB |
| *Size of the entity is large* | CIFAR-10 |
| *Object has a spotted pattern* | ImageNet-100 |

For all experiments related to Table 4, Table 5, we used a standardized textual format: *"Image of a person with (*1) and (2)."* where (*1) and (*2) correspond to the target class and the correlated attribute, respectively. Further experiments using VLM-based textual description in Table 6 for the target attributes Wearing Earrings, Bangs, Big Nose, Heavy Makeup use textual description as *"Person is wearing glasses"*, *"Image of a male person"*, *"Person has long hair"*, and *"Person is wearing glasses"* respectively. For metadata, we used the same format as the Table 4.

## I.6 Dataset Usage Details

Table 15 gives the details about the utilization of training, validation, and test data in our method. The training data, which is biased, is used to fit the model and learn underlying patterns, though it may introduce distributional or simplicity bias. The validation data is used for identifying and analyzing biases present in the trained model. It is also used to generate textual descriptions of bias and hyperparameter tuning. For group-imbalanced datasets such as Waterbirds, we use the standard unbiased validation set provided with the dataset. For standard benchmark datasets, including Awa2, GTSRB, CIFAR-10, and ImageNet-100, where no explicit bias definition exists, we split the original training data into 80% training and 20% validation sets.

For the experiments reported in Table 4 and 5 on datasets with strong spurious correlations, the training data is constructed by subsampling the training set of the original Celeba dataset following the procedure specified in Eyuboglu et al. (2022) to induce spurious correlations. In these settings, we use the original validation split, and additionally sample 20% (relative to the training size) from this validation set for the experiment. Finally, the test data is reserved for the final inference. We have used the test set provided by the original dataset for evaluation.

Table 15: Roles of training, validation, and test datasets with respect to bias and model evaluation.

| Dataset | Purpose | Usage Description |
|---|---|---|
| **Training Data** | Model learning and parameter optimization | Used to train the model; may contain distributional or sampling bias influencing learned representations. |
| **Validation Data** | Bias identification and tuning | Used to detect model bias, hyperparameter tuning, and generate textual descriptions of bias. |
| **Test Data** | Final evaluation and inference | Used only after training and used to assess the model's generalization and unbiased performance. |

## J Methods for Handling Spurious Features

Table 16 outlines the key capabilities and limitations of existing methods relative to ours. While data augmentation techniques (Srivastava et al., 2020; Puli et al., 2022; Yao et al., 2022; Zeng et al., 2020; Wu et al., 2023b; Nam et al., 2022) are widely adopted, they often require external data, which can conflict with

privacy and regulatory constraints (Zhao, 2022; Lim & Oh, 2025; Lonzetta & Hayajneh, 2021; Gstrein & Beaulieu, 2022). Moreover, without appropriate supervision, they risk introducing new spurious features or being vulnerable to data poisoning attacks (Fan et al., 2022; Kumar et al., 2018).

In contrast, methods such as group annotation-based optimization (e.g., gDRO (Sagawa et al., 2020)), loss reweighting techniques (e.g., JTT (Liu et al., 2021)), and final-layer fine-tuning (Kirichenko et al., 2023; LaBonte et al., 2023) do not pose privacy risks. However, in safety-critical applications where models must satisfy stability guarantees (Valentin, 2024; Liu et al., 2024; Zühlke & Kudenko, 2025), these methods can compromise robustness, especially when models are required to ensure Lipschitz continuity for certification. Specifically, they are susceptible to targeted attacks (Hartnett et al., 2019; Li et al., 2025; Neerudu et al., 2023; Holtz et al., 2022), particularly when the training procedure heavily relies on a small subset of influential examples (Cohen et al., 2020; Kumar et al., 2018) used for fine-tuning or reweighting loss values.

Group-balancing techniques (Chaudhuri et al., 2023; Idrissi et al., 2022) partially address these challenges, but often over-prune majority groups. In contrast, our method supports budget-constrained, targeted sample removal, ensuring only detrimental examples are excluded during training.

Furthermore, many of these methods (Sagawa et al., 2020; Chaudhuri et al., 2023; Idrissi et al., 2022) rely on manual group annotations of the training dataset. As spurious features (Lesort, 2023) evolve post-deployment, maintaining robustness would require repeated manual annotation cycles. In contrast, our approach eliminates the need for group labels for the training dataset and leverages textual descriptions of bias to guide targeted data removal. The use of a textual description of the bias and the proposed metric learning approach provides a zero-shot approach (Abdelfattah et al., 2023; Pan et al., 2024) to approximate the underlying group structure without having any annotation overhead. This design also allows integration of feedback from subject-matter experts, making the process more adaptive and practical.

Table 16: Comparison of methods across regulatory and robustness capabilities.

| Method | Regulatory Restrictions | Supports Textual Descriptions | No Group Annotation in Training Data | Privacy | Prevent Over pruning of Majority Group | Robust to Adv-Attacks |
|---|---|---|---|---|---|---|
| Data Augmentation | ✗ | ✗ | ✓ | ✗ | ✓ | ✗ |
| Group-Annotation based Optimization | ✗ | ✗ | ✗ | ✓ | ✓ | - |
| Reweighting Loss/Data | ✗ | ✗ | ✓ | ✓ | ✓ | ✗ |
| Last Layer Fine-Tuning | ✗ | ✗ | ✓ | ✓ | ✓ | ✗ |
| Group Balancing Method | ✓ | ✗ | ✗ | ✓ | ✗ | ✓ |
| **Ours** | ✓ | ✓ | ✓ | ✓ | ✓ | ✓ |

## K Stability across epochs

We have further compared our method to analyze the training stability achieved by our method in comparison to the best-performing optimization-based method (gDRO) in Table 10. As per the result in Table 17, the subset selected by our method provides better stability across training.

Table 17: Worst-group accuracy across epochs for gDRO and Ours.

| | ep 10 | ep 20 | ep 30 | ep 40 | ep 50 |
|---|---|---|---|---|---|
| gDRO | **0.696** | 0.618 | 0.606 | 0.609 | 0.598 |
| **Ours** | 0.524 | **0.681** | **0.726** | **0.737** | **0.742** |

## L Architecture-based Ablation on Worst Group Accuracy and Average Accuracy

We further evaluate our method on the WaterBirds dataset across different architectures, including ResNet-18, VGG16, VGG13, AlexNet, and ConvNet. Pham et al. (2021) shows that the random initial weights can significantly impact the worst group performance of a model, especially in smaller networks. To replicate this setting, we tested our method under extreme conditions, maintaining consistency in textual instructions and using a single run with the same random seed across all baselines.

As shown in Table 19 and Table 18, In comparison with the complete data setting our method achieves an improvement of 5.0%, 1.9%, 3.6%, 3.1%, 12.6% in worst-group accuracy for VGG16, VGG13, Convnet, ResNet18 and AlexNet architecture and an improvement of 5.2%, 7.1% 4.9% and 7.1% for VGG16, ConvNet, ResNet18, and AlexNet in average accuracy respectively.

Furthermore, compared to Trak, our method achieves an improvement of 1.1%, 2.4%, and 4.8% in average group performance for ConvNet, ResNet18, and AlexNet, respectively. Additionally, enhancements of 5.0%, 1.4%, 7.8%, 4.5%, and 12.9% in worst-group performance were observed for VGG16, VGG13, ConvNet, ResNet18, and AlexNet.

Table 18: Architecture ablation on waterbirds (Best Worst Group Accuracy)

| Model | Original | Random | IF | TracIN | EWC | Trak | Ours |
|---|---|---|---|---|---|---|---|
| VGG16 | 0.053 | 0.050 | 0.064 | 0.064 | 0.062 | 0.053 | **0.103** |
| VGG13 | 0.048 | 0.053 | **0.087** | 0.030 | 0.065 | 0.053 | 0.067 |
| ConvNet | 0.090 | 0.034 | 0.064 | 0.033 | 0.053 | 0.048 | **0.126** |
| ResNet18 | 0.050 | 0.064 | 0.067 | 0.017 | 0.048 | 0.036 | **0.081** |
| AlexNet | 0.050 | 0.048 | 0.107 | 0.031 | 0.042 | 0.047 | **0.176** |

Table 19: Architecture ablation on waterbirds (Best Average Accuracy)

| Model | Original | Random | IF | TracIN | EWC | Trak | Ours |
|---|---|---|---|---|---|---|---|
| VGG16 | 0.640 | 0.669 | 0.657 | 0.640 | 0.683 | 0.686 | **0.692** |
| VGG13 | 0.655 | 0.640 | 0.610 | 0.668 | 0.660 | **0.669** | 0.662 |
| ConvNet | 0.654 | 0.705 | 0.640 | 0.721 | 0.711 | 0.714 | **0.725** |
| ResNet18 | 0.641 | 0.604 | 0.600 | **0.694** | 0.623 | 0.666 | 0.690 |
| AlexNet | 0.644 | 0.650 | 0.586 | 0.693 | 0.658 | 0.667 | **0.715** |

## M   Experiment on Vision Transformer

Existing data attribution methods typically compute gradients over all model parameters, which often causes memory issues for large models like Vision Transformers. To address this, we follow recent works (Kokhlikyan et al., 2020; Pruthi et al., 2020) and calculated the gradients only for the final feature layer for both Trak and our method. However, this adaptation was incompatible with other baselines.

The results on Waterbirds for both methods are shown in Table 20.

Table 20: Best Average Accuracy and Best Worst Group performance analysis of our method in comparison with Trak and Original training of vision transformer with entire dataset.

| | Average Accuracy | Worst Group Accuracy |
|---|---|---|
| original | 0.601/0.000 | 0.104/0.000 |
| Trak | **0.644/0.020** | 0.0740/0.014 |
| ours | 0.640/0.027 | **0.1671/ 0.014** |

## N   Relative Comparison with the Baselines

For experiments related to Table 8, we have provided a comparison of the relative performance improvement achieved by our method against other baselines over the complete training data setting. As shown in Table 21 and Table 22, our method, on average, outperforms other baselines in terms of best average accuracy and best worst group accuracy.

Table 21: Relative improvement in Best Average Accuracy (%) achieved by our method and other baselines compared to the complete data setting(Original). The results represent the mean scores from three independent runs, with the best-performing values highlighted in **bold.**

| Target Attribute | Spurious Attribute | Random | EWC | IF | TracIN | Trak | Ours |
|---|---|---|---|---|---|---|---|
| Bangs | Black Hair | -0.03 | 0.37 | 0.36 | -0.07 | 0.16 | **1.05** |
| Big Nose | Wearing Necklace | -0.07 | 1.66 | 1.10 | -0.17 | **2.23** | 2.17 |
| Heavy Makeup | Straight Hair | -0.44 | -1.05 | 0.95 | -2.83 | -0.44 | **2.18** |
| Wearing Earrings | Bags Under Eyes | 0.3 | -0.2 | -1.17 | -0.57 | 0.1 | **0.73** |

Table 22: Relative improvement in Best Worst Group Accuracy (%) achieved by our method and other baselines compared to the complete data setting(Original). The results represent the mean scores from three independent runs, with the best-performing values highlighted in **bold.**

| Target Attribute | Spurious Attribute | Random | EWC | IF | TracIN | Trak | Ours |
|---|---|---|---|---|---|---|---|
| Bangs | Black Hair | 6.25 | **15.62** | 6.66 | 7.05 | 4.77 | 12.58 |
| Big Nose | Wearing Necklace | 4.4 | -3.57 | 5.10 | 4.68 | -4.65 | **22.08** |
| Heavy Makeup | Straight Hair | 1.84 | 5.95 | 4.30 | -1.24 | 3.55 | **6.54** |
| Wearing Earrings | Bags Under Eyes | 5.93 | 1.53 | 13.54 | -3.46 | -3.23 | **24.17** |

## O  Sensitivity Analysis

Table 23 and Table 24 show the sensitivity of our proposed method on different hyperparameter values.

Table 23: Sensitivity analysis of the average accuracy of our method on the WaterBirds dataset for hyperparameters like the barrier constant $(C)$, the matrix rank $(t)$ (shown by rows), and the minimum weight fraction ($\beta$, shown by columns).

| Barrier $(C)$ | Rank $(t)$ | 0.6 | 0.7 | 0.75 | 0.8 | 0.85 | 0.9 |
|---|---|---|---|---|---|---|---|
|  | 40 | 0.657 | 0.619 | 0.648 | 0.673 | 0.650 | 0.640 |
| 5 | 50 | 0.673 | 0.642 | 0.621 | 0.618 | 0.618 | 0.601 |
|  | 100 | 0.670 | 0.678 | 0.650 | 0.602 | 0.632 | 0.631 |
|  | 40 | 0.690 | 0.671 | 0.615 | 0.634 | 0.621 | 0.650 |
| 10 | 50 | 0.633 | 0.679 | 0.639 | 0.657 | 0.629 | 0.609 |
|  | 100 | 0.653 | 0.663 | 0.602 | 0.642 | 0.660 | – |

Table 24: Sensitivity analysis of the worst group accuracy of our method on the WaterBirds dataset for hyperparameters like the barrier constant $(C)$, the matrix rank $(t)$ (shown by rows), and the minimum weight fraction ($\beta$, shown by columns).

| Barrier $(C)$ | rank $(t)$ | 0.6 | 0.7 | 0.75 | 0.8 | 0.85 | 0.9 |
|---|---|---|---|---|---|---|---|
|  | 40 | 0.037 | 0.051 | 0.042 | 0.020 | 0.050 | 0.041 |
| 5 | 50 | 0.033 | 0.042 | 0.055 | 0.048 | 0.056 | 0.065 |
|  | 100 | 0.020 | 0.036 | 0.041 | 0.081 | 0.041 | 0.050 |
|  | 40 | 0.009 | 0.037 | 0.056 | 0.051 | 0.044 | 0.030 |
| 10 | 50 | 0.044 | 0.023 | 0.053 | 0.031 | 0.051 | 0.061 |
|  | 100 | 0.045 | 0.031 | 0.065 | 0.034 | 0.034 | – |

## P   Performance Analysis on Different Subset Size

To further analyze model performance across different subset sizes, we conducted an ablation study where the best hyperparameters were kept fixed while varying the proportion of removed training data for waterbirds. The results are summarized in Table 25.

Table 25: Sensitivity analysis of the worst group accuracy and average accuracy of our method on the WaterBirds dataset for different subset sizes.

| Metrics | 3% | 5% | 15% | 25% |
|---|---|---|---|---|
| Average Accuracy | 0.69 | 0.645 | 0.682 | 0.712 |
| Worst group Accuracy | 0.081 | 0.041 | 0.037 | 0.002 |

## Q   Influence of Validation Data Size

To further analyze the influence of the validation dataset, we have conducted an additional ablation to study how reliably our method identifies spurious features as the size of the validation set varies. Following the same setup as in Table 6 of the main draft and Appendix I.2, we consider training datasets containing spurious features constructed according to Eyuboglu et al. (2022).

Specifically, for a binary classification task (presence vs. absence of the target) with a known spurious feature in the training data, we measure how the score ($\tau$, Ref. Equation 1) associated with spurious features varies with validation data size. As per the result in Table 26, across all validation sizes, the feature with the highest $\tau$ consistently corresponds to the same spurious attribute associated with the training data as per Johnson et al. (2023). Consequently, the generated textual description of the underlying bias remains unchanged.

Table 26: Values of $\tau$ for different validation data sizes (ratio w.r.t. training data)

| Target | Spurious Attribute | 0.1 | 0.2 | 0.4 |
|---|---|---|---|---|
| bangs | black hair | 0.038 | 0.048 | 0.055 |

## R   Performance on Bigger Training Dataset

To further analyze the performance of our method on larger datasets, we conducted additional experiments on training data with sizes comparable to ImageNet-100, consisting of 98,000 training samples and 19,600 validation samples. We follow the experimental setup of Eyuboglu et al. (2022), and introduce more pronounced spurious correlations (approximately 0.1), consistent with the settings used in Tables 6. As shown in Table 27, when spurious features are more prominent, our method achieves noticeable improvements in both average accuracy and worst-group accuracy.

Table 27: Average and Worst-Group Accuracy for bigger dataset

| Target | Spurious Attribute | Original Avg Acc | Ours Avg Acc | Original Worst-Group Acc | Ours Worst-Group Acc |
|---|---|---|---|---|---|
| bangs | black hair | 0.944 | **0.953** | 0.686 | **0.782** |
| heavy makeup | straight hair | 0.888 | **0.894** | 0.790 | **0.821** |

## S   Additional Experiments on Causal vs Spurious Features

To substantiate our claim regarding the undervaluation of detrimental attribution scores, we conducted additional experiments on causal and spurious features following a similar setup described in Section 4.8 for the dataset used in Table 7. As shown in Tables 28, 29, and 30, the mean detrimental attribution score for spurious features is consistently lower than that for causal features. These empirical results closely align with our theoretical analysis.

Table 28: Classifier (Smiling vs Not Smiling) — Statistics for detrimental attribution score. (*** $p < 0.001$, ** $p < 0.01$, * $p < 0.05$; n.s. = not significant.)

| Attribute | Mean | Std | P-value vs (spurious) | Significance |
|---|---|---|---|---|
| Wearing Lipstick (spurious) | 0.499 | 0.041 | — | — |
| high cheekbones | 0.504 | 0.032 | 2.0e-07 | *** |
| mouth slightly open | 0.503 | 0.033 | 3.0e-05 | *** |
| narrow eyes | 0.501 | 0.038 | 0.21 | n.s. |

Table 29: Classifier (Young vs Old) — Statistics for detrimental attribution score. (*** $p < 0.001$, ** $p < 0.01$, * $p < 0.05$; n.s. = not significant.)

| Attribute | Mean | Std | P-value vs (spurious) | Significance |
|---|---|---|---|---|
| smiling (spurious) | 0.442 | 0.047 | — | — |
| bald | 0.452 | 0.070 | 0.004 | ** |
| receding hairline | 0.451 | 0.074 | 6.0e-06 | *** |

Table 30: Classifier (Gray Hair vs Not Gray Hair) — Statistics for detrimental attribution score. (*** $p < 0.001$, ** $p < 0.01$, * $p < 0.05$; n.s. = not significant.)

| Attribute | Mean | Std | P-value vs (spurious) | Significance |
|---|---|---|---|---|
| eyeglasses (spurious) | 0.458 | 0.074 | — | — |
| old | 0.464 | 0.054 | 0.015 | * |
| receding hairline | 0.466 | 0.053 | 0.022 | * |

## T   Time Taken for Subset Selection

In Figure T, we compare the time taken by our method in comparison with other baselines to select a subset of 1200 images from 60,000 images of CIFAR-10 for the instruction mentioned in Table 14. Since our method uses the attribution scores generated by Trak and improves upon it. The time taken by our method is slightly longer than Trak.

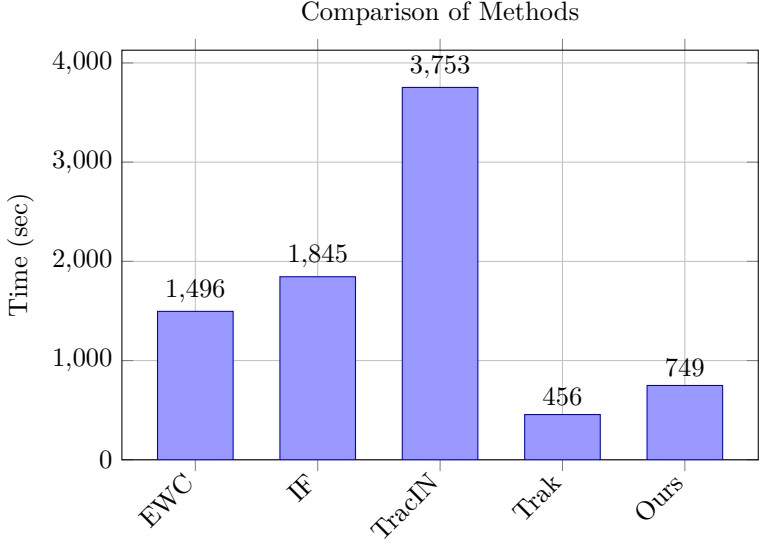

Figure 2: Comparison of time taken to select a subset of 1200 samples from a training dataset of 60,000 images of CIFAR-10 by different baselines and (Ours) for a given textual instruction.

# U   Memory Consumption and Other Training Overhead

The Table 31 reports GPU and RAM usage of our method compared to other baselines, using the same setup described in Appendix T.

As shown, our method introduces only a marginal computational overhead over Trak, which we use for computing data attribution scores. It is to be noted that, while Trak is more memory-intensive, it produces better linear datamodeling score (LDS) scores than other baselines (Park et al., 2023).

Table 31: GPU and RAM utilization (in MB) of our method compared to baseline approaches.

| Method | GPU Memory (MB) | RAM Usage (MB) |
|---|---|---|
| IF | 27,749 | 10,578 |
| EWC | 13,221 | 10,520 |
| TracIN | 44,087 | 9,629 |
| **Trak** | 48,020 | 10,710 |
| **Ours** | **48,525** | **10,722** |

# V   Workflow

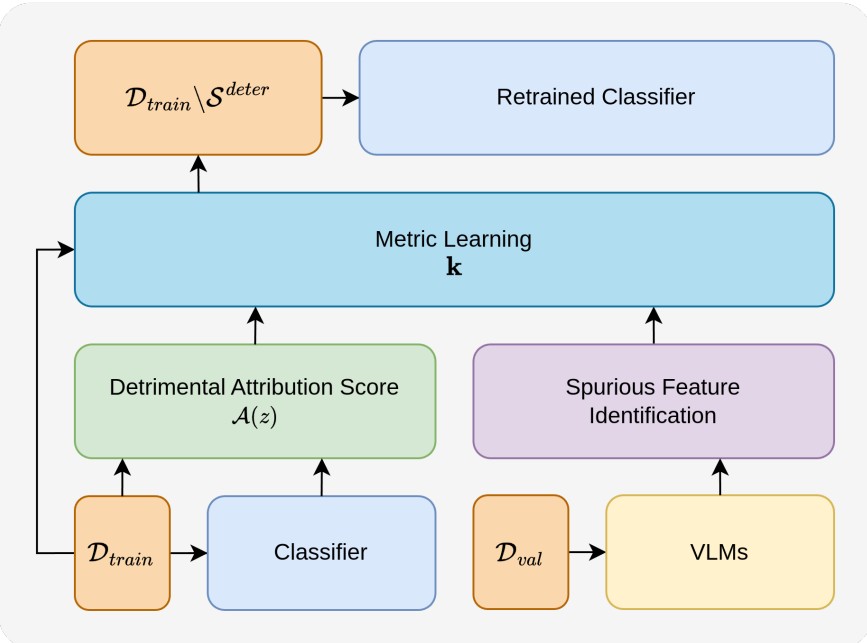

Figure 3: Diagram depicting the workflow of the proposed method

# W   Images

In this section, we have shown the images that have been removed from the training dataset. Figure 4, Figure 5, Figure 6, and Figure 7 show the set of images that have been removed by our method from the training dataset as ($\mathcal{S}^{\text{deter}}$). For WaterBirds, GTSRB, CELEBA, and AWA2-B, respectively.

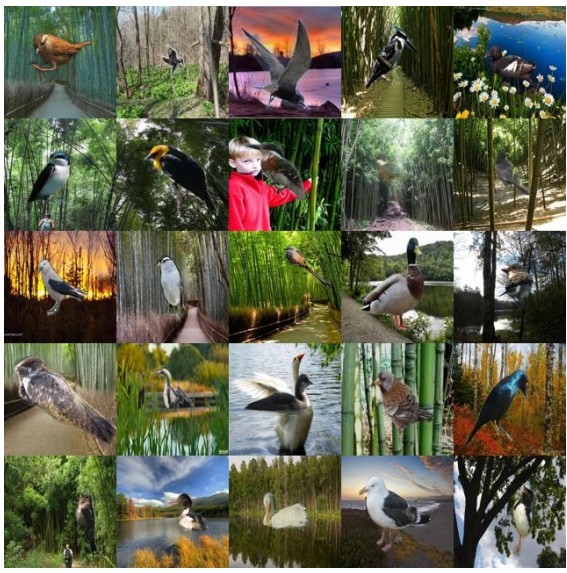

Figure 4: Set of images removed by our method for WaterBirds. The instruction set used for this experiment is *" The surrounding environment in the image is forest floor"*.

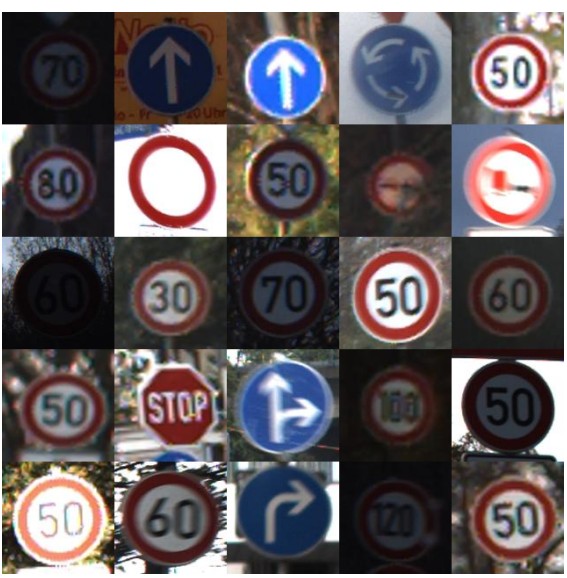

Figure 5: Set of Images removed by our method for GTSRB. The instruction set used for this experiment is *" Shape of sign is round."*

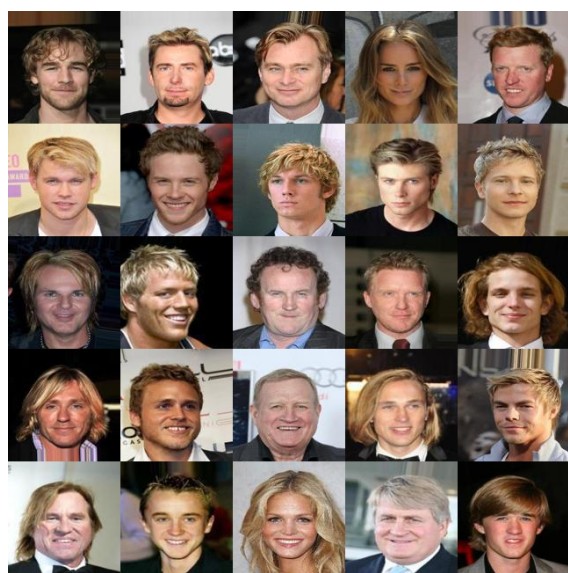

Figure 6: Set of Images removed by our method for CELEBA. The instruction set used for this experiment is *" Image of a male with blond hair"*.

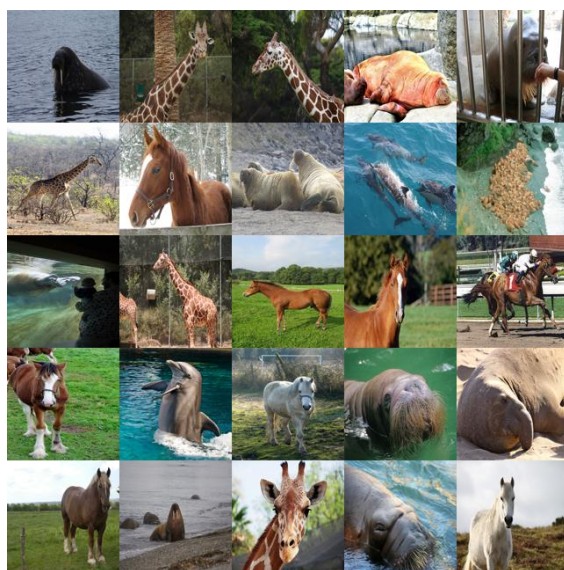

Figure 7: Set of Images removed by our method for Awa2-B. The instruction set used for this experiment is *" The size of the animal is very large."*

# X   Main Results with Standard Deviation

Table 33, Table 32, and Table 34 report the performance of our method and baselines with associated standard deviation.

Table 32: Comparative evaluation of average accuracy of our proposed method (Ours) against baseline approaches across multiple datasets. The results report mean accuracy scores along with standard deviation over three independent runs.

| Dataset | Original | Random | IF | TracIN | EWC | Trak | Ours |
|---|---|---|---|---|---|---|---|
| WaterBirds | $0.638_{0.005}$ | $0.606_{0.004}$ | $0.603_{0.012}$ | $0.652_{0.029}$ | $0.650_{0.022}$ | $0.656_{0.022}$ | $0.681_{0.010}$ |
| AWA2-A | $0.644_{0.009}$ | $0.622_{0.014}$ | $0.644_{0.018}$ | $0.652_{0.009}$ | $0.642_{0.013}$ | $0.638_{0.010}$ | $0.662_{0.014}$ |
| CELEBA | $0.895_{0.004}$ | $0.893_{0.004}$ | $0.890_{0.007}$ | $0.893_{0.007}$ | $0.890_{0.017}$ | $0.898_{0.001}$ | $0.906_{0.003}$ |
| GTSRB | $0.969_{0.006}$ | $0.966_{0.010}$ | $0.973_{0.004}$ | $0.971_{0.001}$ | $0.975_{0.005}$ | $0.971_{0.004}$ | $0.980_{0.002}$ |
| AWA2-B | $0.644_{0.009}$ | $0.622_{0.014}$ | $0.644_{0.018}$ | $0.652_{0.009}$ | $0.642_{0.013}$ | $0.638_{0.010}$ | $0.657_{0.010}$ |
| CIFAR-10 | $0.774_{0.001}$ | $0.787_{0.006}$ | $0.798_{0.002}$ | $0.784_{0.009}$ | $0.789_{0.005}$ | $0.793_{0.003}$ | $0.801_{0.002}$ |
| ImageNet-100 | $0.440_{0.012}$ | $0.436_{0.004}$ | $0.429_{0.005}$ | $0.423_{0.006}$ | $0.424_{0.005}$ | $0.435_{0.010}$ | $0.438_{0.005}$ |

Table 33: Performance comparison of average accuracy across various attribute pairs. Values are reported as $\text{mean}_{\text{std}}$.

| Target | Spurious Attribute | Original | Maj.-Rand | Random | IF | EWC | TracIN | Trak | Ours |
|---|---|---|---|---|---|---|---|---|---|
| arched eyebrows | receding hairline | $0.713_{0.012}$ | $0.740_{0.012}$ | $0.739_{0.012}$ | $0.716_{0.012}$ | $0.724_{0.011}$ | $0.730_{0.002}$ | $0.722_{0.008}$ | $0.736_{0.012}$ |
| attractive | mouth slightly open | $0.628_{0.019}$ | $0.627_{0.014}$ | $0.668_{0.016}$ | $0.640_{0.026}$ | $0.633_{0.019}$ | $0.631_{0.024}$ | $0.658_{0.021}$ | $0.673_{0.010}$ |
| big nose | male | $0.771_{0.012}$ | $0.770_{0.008}$ | $0.770_{0.028}$ | $0.764_{0.032}$ | $0.751_{0.062}$ | $0.745_{0.028}$ | $0.756_{0.025}$ | $0.780_{0.019}$ |
| goatee | bushy eyebrows | $0.946_{0.007}$ | $0.931_{0.009}$ | $0.947_{0.012}$ | $0.938_{0.019}$ | $0.951_{0.005}$ | $0.953_{0.007}$ | $0.949_{0.010}$ | $0.953_{0.006}$ |
| mouth slightly open | smiling | $0.869_{0.012}$ | $0.871_{0.012}$ | $0.877_{0.016}$ | $0.877_{0.014}$ | $0.860_{0.016}$ | $0.876_{0.014}$ | $0.867_{0.009}$ | $0.877_{0.014}$ |
| mouth slightly open | wearing lipstick | $0.820_{0.006}$ | $0.804_{0.018}$ | $0.801_{0.033}$ | $0.829_{0.007}$ | $0.834_{0.011}$ | $0.817_{0.024}$ | $0.801_{0.012}$ | $0.839_{0.016}$ |
| narrow eyes | eyeglasses | $0.840_{0.023}$ | $0.858_{0.012}$ | $0.862_{0.004}$ | $0.856_{0.011}$ | $0.858_{0.004}$ | $0.860_{0.013}$ | $0.855_{0.012}$ | $0.862_{0.008}$ |
| pointy nose | mouth slightly open | $0.690_{0.029}$ | $0.714_{0.008}$ | $0.676_{0.025}$ | $0.689_{0.002}$ | $0.695_{0.011}$ | $0.709_{0.015}$ | $0.694_{0.008}$ | $0.698_{0.018}$ |
| receding hairline | rosy cheeks | $0.921_{0.005}$ | $0.909_{0.012}$ | $0.920_{0.007}$ | $0.921_{0.004}$ | $0.920_{0.001}$ | $0.916_{0.009}$ | $0.911_{0.012}$ | $0.930_{0.009}$ |
| male | pointy nose | $0.919_{0.013}$ | $0.931_{0.006}$ | $0.907_{0.017}$ | $0.909_{0.010}$ | $0.911_{0.008}$ | $0.906_{0.015}$ | $0.915_{0.014}$ | $0.921_{0.009}$ |

Table 34: Worst-group accuracy across various spurious Attributes. Values are reported as $\text{mean}_{\text{std}}$.

| Target | Spurious Attribute | Original | Maj.-Rand | Random | IF | EWC | TracIN | Trak | Ours |
|---|---|---|---|---|---|---|---|---|---|
| arched eyebrows | receding hairline | $0.187_{0.218}$ | $0.314_{0.208}$ | $0.113_{0.160}$ | $0.247_{0.141}$ | $0.262_{0.092}$ | $0.196_{0.097}$ | $0.099_{0.097}$ | $0.354_{0.131}$ |
| attractive | mouth slightly open | $0.213_{0.130}$ | $0.242_{0.064}$ | $0.347_{0.079}$ | $0.266_{0.097}$ | $0.241_{0.108}$ | $0.205_{0.051}$ | $0.392_{0.056}$ | $0.407_{0.018}$ |
| big nose | male | $0.131_{0.048}$ | $0.076_{0.046}$ | $0.096_{0.069}$ | $0.143_{0.059}$ | $0.092_{0.119}$ | $0.113_{0.045}$ | $0.172_{0.041}$ | $0.221_{0.120}$ |
| goatee | bushy eyebrows | $0.432_{0.100}$ | $0.493_{0.085}$ | $0.287_{0.090}$ | $0.438_{0.148}$ | $0.439_{0.092}$ | $0.387_{0.065}$ | $0.278_{0.082}$ | $0.548_{0.050}$ |
| mouth slightly open | smiling | $0.524_{0.070}$ | $0.415_{0.137}$ | $0.552_{0.115}$ | $0.419_{0.107}$ | $0.441_{0.111}$ | $0.487_{0.020}$ | $0.433_{0.115}$ | $0.489_{0.077}$ |
| mouth slightly open | wearing lipstick | $0.555_{0.028}$ | $0.471_{0.043}$ | $0.557_{0.041}$ | $0.598_{0.079}$ | $0.594_{0.056}$ | $0.549_{0.049}$ | $0.486_{0.123}$ | $0.612_{0.076}$ |
| narrow eyes | eyeglasses | $0.208_{0.155}$ | $0.052_{0.048}$ | $0.119_{0.085}$ | $0.000_{0.000}$ | $0.092_{0.079}$ | $0.128_{0.121}$ | $0.024_{0.034}$ | $0.151_{0.029}$ |
| pointy nose | mouth slightly open | $0.045_{0.017}$ | $0.044_{0.033}$ | $0.046_{0.018}$ | $0.034_{0.006}$ | $0.028_{0.015}$ | $0.021_{0.004}$ | $0.040_{0.016}$ | $0.084_{0.028}$ |
| receding hairline | rosy cheeks | $0.121_{0.095}$ | $0.228_{0.104}$ | $0.131_{0.095}$ | $0.180_{0.031}$ | $0.241_{0.056}$ | $0.254_{0.146}$ | $0.201_{0.080}$ | $0.296_{0.091}$ |
| male | pointy nose | $0.840_{0.040}$ | $0.882_{0.024}$ | $0.824_{0.072}$ | $0.833_{0.047}$ | $0.861_{0.022}$ | $0.870_{0.026}$ | $0.875_{0.024}$ | $0.903_{0.009}$ |

