# OpenReview forum: "An Efficient Subset Selection Strategy Using Text-Guided Data Attribution to Mitigate Simplicity Bias"
_TMLR — Accepted by TMLR_

### Review · Reviewer_zAZT · 2025-11-22

**Summary Of Contributions:**

The authors propose a two-stage data filtering framework aimed at retaining high–information content samples while identifying and removing spurious or overly simplistic examples. The overall motivation is clear: curating higher-quality training subsets can significantly improve downstream fine-tuning performance, yet this step is often neglected in practice.

In the first stage, the authors rely on metadata or GPT-/Llama-based annotations to estimate how individual datapoints contribute to the model’s overall accuracy. This forms a coarse pre-screening mechanism.

The second stage builds on a more principled data attribution framework, specifically the leave-one-out influence score, which measures the change in a model’s prediction on a validation sample when a given training point is removed. Leveraging NTK-based methods, the authors employ a closed-form estimator to compute these influence scores across multiple model checkpoints. They further extend this by defining a detrimental attribution score, A(z), that captures the effect of removing a training point on the entire validation set.

The paper appropriately highlights a known limitation of attribution scores: due to learning dynamics such as gradient starvation, samples containing spurious features may be assigned deceptively low harmfulness scores. As articulated in Proposition 1, this can lead naïve deletion strategies to remove genuinely informative samples while retaining spurious ones.

To address this, the authors introduce an augmented filtering mechanism that incorporates zero-shot textual bias descriptions with CLIP embeddings. A semantic similarity score is computed between each training example and the textual description of the suspected spurious feature, using a learnable low-rank distance metric. The final removal strategy jointly considers (1) the detrimental attribution score and (2) semantic alignment with spurious features. The learning objective for the metric matrix L balances these two factors through a tunable hyperparameter T.

Overall, the approach combines influence-based data curation with semantic bias identification in a thoughtful and complementary manner.

**Audience:**

Yes

**Audience Explanation:**

This is indeed an important—and often under-emphasized—step in fine-tuning workflows, and the method has the potential to make downstream experiments more efficient and more robust by explicitly targeting spurious correlations. The paper’s direction is promising, though further empirical validation and ablation studies would strengthen the contribution

**Broader Impact Concerns:**

The metadata generated from other VLM/LLM might propagate bias. This is something to be aware of..

**Claims And Evidence:**

Yes

**Claims Explanation:**

- They show clear improvement in multiple metrics
- Show comparison with multiple models eg ResNet18, VGG16, VGG13, AlexNet, and ConvNet (P.S: Paper mentions Appendix L but its actually appendix K
- Shows comparison across  multiple methods establishing a good baseline
- Details on training, hyperparameters, and subset size are provided
- They have shared github link for repro

**Requested Changes:**

It would strengthen the paper to have a short section on potential Responsible AI implications

---

> ### Author Response · Authors · 2025-12-20
> **Response to Reviewer - zAZT**
>
> Thank you for reviewing our work and for the supportive feedback. We appreciate that you recognized the impact of our approach, and we address your questions as below.
>
> ### **R3.1 Bias associated with VLM**
> We would like to highlight that the current state-of-the-art approaches [1] for identifying spurious features and generating corresponding textual descriptions rely on LLM- or VLM-based methods. In our framework, VLMs are used only to generate high-level textual descriptions of bias. When suitable metadata is available, or when a subject-matter expert can directly provide such descriptions, the use of VLMs can be entirely avoided, and hence can prevent any influence of such a model.
>
> Furthermore, while our analysis demonstrates how spurious features can influence attribution scores, the attribution score can be resilient to potential biases introduced by the VLM. Hence, by prioritizing attribution scores over textual descriptions through the weighting parameter $\mathcal{T}$, the influence of the VLM can be effectively controlled and minimized.
>
> We will incorporate this discussion into the revised draft as part of responsible AI.
>
> ### References
>
> [1] HiBug: On Human-Interpretable Model Debug, Neurips 2023
>
>
> ### **R3.2 Adding discussion on responsible AI implications**
>
> Thank you for the suggestion. We have added a new section on responsible AI in the main draft. Please refer to **Appendix F** for more details.
>
>
> ### **R3.3 Additional Ablation and Empirical Evidence**
>
> Thank you for the suggestion. We have included additional experiments to demonstrate the influence of spurious features on the attribution score, along with a new ablation studying the effect of validation data size and large training datasets. Please refer to our responses to **R1.4**, **R1.5** and **R1.8** for Reviewer **X9wM** for details.
>
>
> ### **R3.4 Appendix name reference**
> Thank you for pointing this out. In the old draft, Appendix L contains the ablation studies for the vision transformer, while Appendix K presents the corresponding ablations for CNN architectures. We will revise the draft to make this distinction explicit.

---

### Review · Reviewer_Axbk · 2025-12-11

**Summary Of Contributions:**

The paper aims to address the problem of simplicity bias in deep learning, where models rely on spurious features rather than predictive ones. The authors argue that existing data attribution methods ignore training samples containing spurious features due to gradient starvation. To mitigate this, the paper proposes a two-stage text-guided data subset selection strategy. First, spurious features are identified using validation data annotated by VLMs. Then, a metric learning approach that combines CLIP embeddings with data attribution scores is used to identify and remove samples that are semantically aligned with the bias and detrimental to generalization. The authors conduct experiments on a wide range of benchmarks and show improvements in average and worst-group accuracy compared to baselines.

**Contributions**
- The authors provide a theoretical proposition suggesting that in the presence of gradient starvation, NTK-based attribution scores for samples with spurious features are systematically undervalued compared to those with predictive features.
- The paper introduces a novel pipeline that utilizes VLM-generated annotations to detect spurious features in validation sets and subsequently uses these textual descriptions to guide a metric learning objective. This objective is used to delete a subset of data based on semantic similarity to the bias and detrimental attribution scores.
- The proposed method is shown to outperform existing data attribution baselines in terms of average and worst-group accuracy across multiple datasets.

**Strengths**
- The approach of bridging the gap between data attribution and semantic understanding with VLMs is novel. The integration of VLMs to annotate validation data when metadata is unavailable offers a practical solution for in-the-wild datasets where group labels are missing.
- The connection drawn between gradient starvation and the failure of NTK-based attribution is valuable. Proposition 1 formalizes the claim that attribution scores for samples with dominant spurious features can be undervalued, serving as a strong motivation.
- The results demonstrate consistent improvements over relevant baselines across a wide variety of datasets and include meaningful ablations.

**Weaknesses**
- The proof for Proposition 2 relies on strong, simplifying assumptions that may not hold generally. In Eq. 24, the authors assume that for two data points $z_i$ and $z_j$, the projection of the gradient onto the dominant feature space satisfies $u_1^i = u_2^j$ to prove the inequality. This is a strong assumption for a "real-world" neural network argument.
- The definition of "Detrimental Attribution Score" is defined as the negative sum of attribution scores, but in the discussion of gradient starvation, the text states that spurious features result in lower scores. This seems to be conflicting with each other.
- Tables 1 and 5 compare primarily against data attribution/pruning methods. However, the problem statement of mitigating spurious bias is a domain dominated by robust training methods like gDRO or JTT. In Table 16, gDRO seems to perform better than the proposed method. While the authors argue that their method does not require group annotations during training, they do require validation annotations. The distinction in performance vs. supervision requirements should be more transparent in the main text.

**Audience:**

Yes

**Audience Explanation:**

The finding that standard attribution methods might fail in the presence of strong spurious features due to gradient starvation is of high interest at the intersection of data-centric AI, interpretability, and robustness.

**Claims And Evidence:**

Yes

**Claims Explanation:**

- The empirical claims regarding improvement over attribution baselines are well-supported.
- The claim that the method mitigates simplicity bias is supported.
- The theoretical claim that NTK attribution scores are undervalued for spurious features is supported by a derivation in Appendix E, but as noted above, this derivation relies on specific symmetry assumptions.

**Requested Changes:**

- In Appendix E, discuss what happens when the symmetry of $u_1^i = u_2^j$ is broken. If this is a simplified illustrative case, explicitly frame it as such in the main text (Proposition 1) rather than a general proof.
- Move the summary of the comparison with Group DRO and JTT to the main text.

---

> ### Author Response · Authors · 2025-12-20
> **Response to Reviewer -  Axbk**
>
> We thank reviewer Axbk for their feedback, and we are glad that they feel that our findings are valuable for data-centric AI. We address their concerns below.
> ### **R2.1 dominant feature satisfying $u^i_1 = u^j_2$**
> Thank you for bringing this up. As defined in **Definition 2** (Appendix E) and [1], $u_1^i$ and $u_2^j$ quantify the contribution of feature $f_1$ (spurious) and $f_2$ (causal) to samples $z_i$ and $z_j$, respectively, as formalized in Proposition 1 and Definition 3.
>
> In Proposition 1, our objective is to illustrate the effect of gradient starvation. Specifically, we show that for two samples with equal feature contribution, a sample ($z_i$) dominated by a feature ($f_1$) that induces gradient starvation can receive a lower detrimental attribution score than a sample ($z_j$) dominated by a causal feature. Consequently, methods that rely solely on attribution scores may inadvertently prioritize the removal of samples containing causal information.
>
> To isolate and analyze this effect, we assume $u_1^i = u_2^j$, **representing equal representation of the features $f_1$ and $f_2$ in sample $z_i$ and $z_j$**. This simplifying assumption in our theoretical analysis allows us to clearly demonstrate how gradient starvation can bias attribution-based subset selection. In the final draft, we will explicitly clarify the role and interpretation of these quantities and the motivation behind this assumption.
>
> #### References
>
> [1]  Gradient starvation: A learning proclivity in neural networks., NEURIPS 2021
>
> ### **R2.2 Real-World Scenarios When $u^i_1 \neq u^j_2$**
> As shown in our analysis in Appendix E, for dominant features $f_1$ and $f_2$ and samples $z_i$ and $z_j$, the detrimental attribution score is largely influenced by the ratios $$\frac{u_1^i}{s_{11}} \quad \text{and} \quad \frac{u_2^j}{s_{22}}.$$ where "$u$" captures the contribution of the feature and "$s$" the overall strength of the feature.
> Since, for scenarios when $s_{11} > s_{22}$, feature $f_1$ induces gradient starvation of feature $f_2$ and for equally informative features, this leads to $$\frac{u_1^i}{s_{11}} < \frac{u_2^j}{s_{22}},$$ resulting in a lower attribution score for the sample dominated by the spurious feature.
>
> In common real-world settings, data samples often predominantly contain either causal or spurious features (i.e., $u_2^i \approx 0$ and $u_1^j \approx 0$), and for some training samples the contribution of causal features may exceed that of spurious ones (i.e., $u_1^i < u_2^j$). Even in this case, the inequality
> $$
> \frac{u_1^i}{s_{11}} < \frac{u_2^j}{s_{22}}
> $$
> continues to hold when $s_{11} > s_{22}$, implying that samples dominated by **causal features** may be preferentially removed. Intuitively, due to gradient starvation, the model overestimates the relevance of spurious features and incorrectly treats samples containing causal information as detrimental.
>
> In more general settings, sample selection depends not only on feature contributions but also on the relative strength of features ($s_{11}$ and $s_{22}$). We plan to study this interaction in greater detail as part of future work; we will add this discussion in the main draft.
> ### **R2.3 text mention low score**
> Thank you for pointing out this typo; we will fix it in the draft. As stated in the proof, we meant the low detrimental attribution score.
> ### **R2.4 Distinction with other baselines and need of supervision for validation data**
>
> We would like to clarify that although our method uses annotations for the **validation dataset**, either available through metadata or generated using the help of LLM or VLM, it introduces several unique features.
>
> - Unlike methods such as GroupDRO and JTT, which are primarily model-centric and aim to improve robustness by changing the training objective or reweighting the samples during training, our approach is data-centric and focuses primarily on identifying and improving the quality of the training dataset. For applications where changes to the model training process are prohibited, this could be useful See Appendix J for further details and Appendix K for ablation on stability.
>
>  - Validation annotations are used only to identify spurious features and construct a textual description of the underlying bias. Unlike other methods, the identification of detrimental training samples relies on this textual description of bias, not on raw per-sample labels. This design enables practical human-in-the-loop scenarios, where a domain expert can directly provide a textual description of the bias without exhaustively annotating data.
>
> To the best of our knowledge, this is the first work that addresses bias in training data using textual bias descriptions rather than explicit sample-level supervision.  However, since current work uses validation data to generate a textual description of the bias, we will mention it in the main draft.
>
> ### **R2.5 summary to main text**
> We have added section 4.9 as per the suggestion of the reviewer.

---

### Review · Reviewer_X9wM · 2025-12-13

**Summary Of Contributions:**

The authors propose a data-centric method to mitigate simplicity bias (reliance on spurious correlations) in image models by identifying and removing detrimental training samples without requiring full group annotations. The authors theoretically demonstrate that standard NTK-based data attribution methods (like Trak) can systematically undervalue the influence of samples containing spurious features, and as a result the naïve pruning based on attribution scores may remove good data compared to spurious data. The authors then propose a pipeline to use an unbiased validation set to identify spurious attribute-value pairs, generate a textual description of the bias, then employ a metric learning objective (CLIP embedding + attribution scores) to select samples and remove them.

Strengths:
1.	The paper is clearly written, the modules (identify, score, semantically focused deletion) are well explained.
2.	The method does not require group annotations of training data, only a small validation set.
3.	The breadth of the empirical results covers multiple datasets and baselines. Ablations against only attribution/CLIP also show complementary gains when combined.

Weakness:
1.	The method's performance gains are very modest, and the behaviors are somewhat inconsistent across different benchmarks.
2.	The method relies heavily on a clean, unbiased validation set. It is not clear how this is done.
3.	The method relies on LLM for attribute generation + annotation where the cost can be relatively high, and reproducibility may be limited.
4.	It appears the method is quite sensitive to different hyperparameters with many tunable parameters.

**Audience:**

Yes

**Audience Explanation:**

Data is a key problem in modern ML, and the community is generally interested in how to best utilize/prune the training data to obtain the best results. The proposed method is practically useful and without expensive manual annotations. While the method is not ground-breaking, the theoretical analysis still provides guidance to researchers (e.g. who use influence functions regularly).

**Claims And Evidence:**

Yes

**Claims Explanation:**

1.	The motivation for why existing attribution methods fail is clear. The theoretical evidence is compelling. The derivation supports the claim that spurious features may result in lower attribution scores.
2.	Empirically, the experiments cover a wide range of datasets and architectures. The baselines are thorough. The ablations are decent.

**Requested Changes:**

1. Since the spurious detection depends on the unbiased validation set, please precisely describe how the unbiased validation split is constructed. Ablations should be provided for the validation set (e.g., size, if it’s noisy).
2. I believe Table 9 should be expanded to multiple datasets/spurious features. In general, consider reporting results on more attribute pairs to establish consistent patterns.
3. The authors should report the deletion rate per experiment, and detail how the hyperparameters (e.g. $\gamma$) are chosen.
4. Could the authors justify whether the improvements warrant the computational overhead?
5. It appears ImageNet-100 had minimal improvements. Can the authors discuss scalability to larger datasets? Is there a scaling limitation?

---

> ### Author Response · Authors · 2025-12-20
> **Response To Reviewer X9wM -- Part (1/4)**
>
> We thank reviewer X9wM for their thorough review and for raising some important questions about the experiment design. We are glad that the reviewer liked our empirical and ablation study. We address all questions below and will incorporate mentioned changes in the main draft.
>
> ### **R1.1 Modest performance gain and inconsistency**
> Thank you for raising this point. We evaluate our method on five types of datasets to assess robustness across diverse real-world conditions:
>
> 1. **Group-imbalanced datasets** (Waterbirds).
> 2. **Manually curated benchmark datasets** (CIFAR-10, ImageNet-100).
> 3. **Datasets with class-level metadata** (Awa2).
> 4. **Datasets affected by environmental spurious features** (GTSRB).
> 5. **Datasets with strong spurious correlations** [1] ( Table 4, 5 for CELEBA).
>
> As noted in prior work [1], spurious features often arise from the data collection process and associated feature imbalances in the training dataset. While previous studies on vision datasets have primarily focused on datasets such as CelebA and Waterbirds, where spurious correlations are particularly pronounced, and have mainly evaluated improvements in worst-group accuracy [2,3], our work takes a broader perspective. To the best of our knowledge, we are the first to systematically analyze the impact of spurious features not only on spurious-prone datasets but also across standard benchmark datasets.
>
>
> Our results show that while the largest gains in average accuracy occur for datasets where spurious features are prominent, we observe consistent improvements in worst-group accuracy and class-level performance across all dataset types, a metric commonly used to assess robustness under distributional shift [4]. Specifically, for benchmark datasets such as Awa2, GTSRB, CIFAR-10, and ImageNet-100, our method improves performance for nearly half of the classes, with average gains ranging from **5% to 17%** (Table 2), while maintaining equivalent or better accuracy for the worst-performing class (Table 3). This indicates that the pruned models remain robust for practical deployment while maintaining comparable or improved average accuracy.
>
> For datasets with strong spurious correlations (Table 5), our method improves **worst-group accuracy by upto 15%** and outperforms **TRAK by 10.6% on average** across datasets, highlighting its effectiveness in settings where simplicity bias is particularly pronounced.
>
> #### References
>
> [1] Domino: Discovering Systematic Errors With Cross-Modal Embeddings, ICLR 2024
>
> [2] Distributionally Robust Neural Networks for Group Shifts, ICLR 2020
>
> [3] Why does Throwing Away Data Improve Worst-Group Error?, ICML 2023
>
> [4] A Survey on Evaluation of Out-of-Distribution Generalization, arxiv 2024
>
>
>
>
> ### **R1.2 Utilization of validation dataset**
>
> In general, for datasets used to study spurious features, we follow the validation protocol provided with the original dataset whenever available. If an unbiased validation split is explicitly provided, we use it for our experiments. When neither an explicit validation set nor a well-defined notion of bias is available, we construct validation data by splitting the training set.
>
>
> As noted in **R1.1**, for group-imbalanced datasets such as Waterbirds, we use the standard unbiased validation set provided with the dataset. For standard benchmark datasets, including Awa2, GTSRB, CIFAR-10, and ImageNet-100, where no explicit bias definition exists, we split the original training data into 80% training and 20% validation sets.
>
> For the experiments reported in Tables 4 and 5 on datasets with strong spurious correlations, the training data is constructed by subsampling the training set of the original CelebA dataset following the procedure specified in [1] to induce high spurious correlations. In these settings, we use the original validation split and additionally sample 20% (relative to the training size) from this validation set for the experiment.
>
> #### References
>
> [1] Domino: Discovering Systematic Errors With Cross-Modal Embeddings, ICLR 2024

---

> ### Author Response · Authors · 2025-12-20
> **Response To Reviewer X9wM -- Part (2/4)**
>
> ### **R1.3 Hyperparameter and its sensitivity**
>
> As discussed in Appendix O, our method tunes only three hyperparameters: the barrier term \(C\), the rank \(t\), and the weight ratio \($\beta$\). For fair comparison, all other hyperparameters are kept constant across datasets and baselines.  We want to highlight that prior works addressing spurious features are sensitive to hyperparameter tuning and often involve tuning a similar or larger set of hyperparameters [1,2].
>
> Based on the ablation results in Tables 23 and 1, for every combination of ( C ) and (t), there exists at least one value of ($\beta$) for which our method outperforms both the original data and the second-best baseline, TRAK. Moreover, the optimal values of \($\beta$\) consistently fall within a narrow range, varying by at most 0.2 across different choices of ( C ) and (t), suggesting that for practical purposes exhaustive tuning may not be required.
>
> We emphasize that ($\beta$) controls the trade-off between the attribution score and the CLIP embeddings, determining the relative importance of these two signals. Since the reliability of attribution scores depends not only on gradient starvation but also on how well features are represented in the data (see R2.2, response to **Reviewer Axbk** ), the observed sensitivity to ($\beta$) aligns with our theoretical analysis. As future work, we plan to study this behavior more deeply and develop methods to automatically select ($\beta$) based on the characteristics of the dataset and attribution statistics.
>
> ####  References
>
> [1] Sagawa et al., Distributionally robust neural networks for group shifts: On the importance of regularization for worst-case generalization. ICLR 2020.
>
> [2] Liu et al., Just train twice: Improving group robustness without training group information. ICML 2021.
>
>
> ### **R1.4  Ablation with validation size**
>
> Since the validation data is used only to generate a textual description of the underlying bias, we conducted an additional ablation to study how reliably our method identifies spurious features as the size of the validation set varies. Following the same setup as in Table 8 of the main draft and Appendix I.2, we consider training datasets containing spurious features constructed according to [1].
>
> Specifically, for a binary classification task (presence vs. absence of the target) with a known spurious feature in the training data, we measure how the score ( $\tau$ Ref equation 1) associated with spurious features varies with validation data size. As per the result in Table A, across all validation sizes, the feature with the highest $\tau$ consistently corresponds to the same spurious attribute associated with the training data as per [2]. Consequently, the generated textual description of the underlying bias remains unchanged. While current work primarily focuses on studying spurious features and attribution scores, we will further study the influence of different characteristics of validation data as part of our future work.
>
> ### Table A: Values of ($\tau$) for different validation data sizes (ratio w.r.t. training data)
>
> | Target | Spurious Attribute | 0.1 | 0.2 | 0.4 |
> |--------|---|-----|-----|-----|
> | bangs  | black hair  | 0.038 | 0.048 | 0.055 |
>
> #### Reference
>
> [1] Domino: Discovering Systematic Errors With Cross-Modal Embeddings, ICLR 2024
>
> [2] Where does my model underperform? a human evaluation of slice discovery algorithms.  AAAI-Conference on Human Computation and Crowdsourcing, 2023

---

> ### Author Response · Authors · 2025-12-20
> **Response To Reviewer X9wM -- Part (3/4)**
>
> ### **R1.5 Expanding Table 9**
>
> As per the suggestion from the reviewer, we have conducted additional experiments on different classification tasks to compare the behavior of causal features versus spurious features, following the same experiment setup as used in Table 9. As per the results shown in Table B,C,D the mean detrimental attribution score for spurious features is consistently lower than that of causal features, which aligns with our theoretical analysis.
>
>
> ### Table B:  Classifier (Smiling vs Not Smiling) — Statistics for detrimental attribution score
> (*** p < 0.001, ** p < 0.01, * p < 0.05; n.s. = not significant.)
>
> | Attribute  | Mean   | Std    | P-value vs (spurious) | Significance |
> |-|-|--------|---|--|
> | Wearing Lipstick (spurious) | 0.499 | 0.041 | —   | —            |
> | high cheekbones  | 0.504 | 0.032 | 2.0e-07               | ***  |
> | mouth slightly open| 0.503 | 0.033 | 3.0e-05               | *** |
> | narrow eyes   | 0.501 | 0.038 | 0.21                   | n.s. |
>
> ### Table C : Classifier( Young vs Old) — Statistics for detrimental attribution score
> | Attribute  | Mean   | Std    | P-value vs (spurious) | Significance |
> |-|--|---|--|-|
> | smiling (spurious)    | 0.442 | 0.047 | —                      | — |
> | bald                  | 0.452 | 0.070 | 0.0043                 | ** |
> | receding hairline     | 0.451 | 0.074 | 6.0e-06               | *** |
>
>
> ### Table D : Classifier (Gray Hair vs Not Gray Hair) — Statistics for detrimental attribution score
>
> | Attribute                  | Mean   | Std     | P-value vs (spurious) | Significance |
> |---|--|---|----|----|
> | eyeglasses (spurious)| 0.458 | 0.074 | —                      | —  |
> | old  | 0.464 | 0.054  | 0.015 | * |
> | receding hairline| 0.466 | 0.053 | 0.022                 | * |
>
> ### **R1.6 Deletion rate and how $\gamma$ is chosen**
> For all datasets reported in Table 1 (Awa2, CELEBA, waterbirds, CIFAR10, ImageNet-100 ), we remove 3% of the training data. Specifically, samples are ranked by $k_i$ , and the threshold ($\gamma$) corresponding to the top 3rd percentile is used for removal.
>
> For all other experiments and ablations, we have followed a similar setup and kept the deletion rate constant across experiments and baselines. Further details on subset size are provided in Appendix I.4, along with a subset-size based ablation study in Appendix P.
>
>
> ### **R1.7 Computational overhead and improvement**
>
> Thank you for raising this point. While our method yields a modest improvement of ~1–3% in average accuracy, the gains in class level and group level performance are consistent. In particular, for biased datasets we observe an average **10.6% improvement in worst-group accuracy** over TRAK across datasets (Table 5) supporting the benefit of the proposed method and in line with the theoretical analysis.  These results indicate that our approach can improve model robustness under distributional shift, which is critical for real-world deployment.
>
> We also note that the primary computational cost arises from annotating the validation dataset, which is used only to generate textual descriptions of bias. When metadata is available or when a subject-matter expert can directly infer these descriptions by inspecting model behavior, this cost can be substantially reduced.
>
> Finally, given the difficulty of collecting perfectly balanced datasets, especially in domains such as healthcare [1,2,3], we believe our method provides a practical tool for better-informed data curation and robustness improvement in real-world applications.
>
> #### References
>
> [1] Domino: Discovering Systematic Errors With Cross-Modal Embeddings, ICLR 2024
>
> [2] Uncovering and Correcting Shortcut Learning in Machine Learning Models for Skin Cancer Diagnosis,MDPI 2022
>
> [3] “Shortcuts” Causing Bias in Radiology Artificial Intelligence: Causes, Evaluation, and Mitigation, PMC, 2023

---

> ### Author Response · Authors · 2025-12-20
> **Response To Reviewer X9wM -- Part (4/4)**
>
> ### **R1.8 performance for large dataset**
>
> We do not believe that our method has any inherent scalability limitations. Instead, the performance improvement may depend on the prominence of spurious features in the training dataset. Datasets such as ImageNet-100 are manually curated and contain relatively balanced class distributions. As per the results in Table 1 for imageNet-100, while the original dataset yields strong performance, our method still outperforms other attribution-based baselines in terms of average accuracy and achieves substantial improvements in class-level performance (Table 2).
>
> To further validate this claim, we conducted additional experiments on training datasets with sizes comparable to ImageNet-100 (approximately 100k samples), following the setup in [1], but with more pronounced spurious correlations (similar to the settings in Tables 4 and 5). As shown in Table E, when spurious features are more prominent, our method yields noticeable improvements in both average accuracy and worst-group accuracy, supporting our claim.
>
> However, we would like to further investigate the influence of spurious features on benchmark dataset like ImageNet-100 as part of our future work.
>
> ### Table E: Average and Worst-Group Accuracy for bigger dataset
>
> | Target        | Spurious Attribute | Original Avg Acc | Ours Avg Acc | Original Worst-Group Acc | Ours Worst-Group Acc |
> |--|--|--|--|--|--|
> | bangs| black hair| 0.944 | **0.953**| 0.686               | **0.782**                |
> | heavy makeup  | straight hair| 0.888 | **0.894** | 0.790   | **0.821**|
> #### References
>
> [1] Domino: Discovering Systematic Errors With Cross-Modal Embeddings, ICLR 2024

---

### Author Response · Authors · 2025-12-20
**General Comment**

We thank all reviewers for their constructive and insightful feedback. Their comments have improved both the clarity and quality of our draft.
Based on the suggestions, we have made the following revisions to the main paper and appendix (highlighted in blue):

- **Proposition 1:** Clarifies the contribution of features on data samples using Definition 3. Reviewer *Axbk*.
- **Section 4.9:** Moved the section on comparison with the robustness-based method in the main draft, following comments from Reviewer *Axbk* .


#### Appendix Updates

- **Definition 3:** Described the contribution of a feature on a sample (*Axbk*).
- **Appendix E.1:** Added real-world implication of the given theoretical analysis, especially when the feature contributions are not equal (*Axbk*).
- **Appendix F** Added discussion on Responsible AI (*zAZT*).
- **Appendix I.6** describes the use of validation dataset (x9wM)
- **Appendix Q:** Added Ablation on validation data size (*X9wM*, *zAZT* ).
- **Appendix R:** Provides a comparison on bigger training data (*X9wM*, *zAZT* ).
- **Appendix S:** Provides empirical support to the theoretical analysis (extends Table 9) (*X9wM*, *zAZT* ).


We have tried to address all major concerns raised by the reviewers. If any clarification is still needed, we are happy to provide it.

---

### Decision · Action_Editor_JxnL · 2026-01-30

**Recommendation:** Accept as is

**Additional Comments:**

Overall, after rebuttal, the authors addressed the most pertinent identified weaknesses, and therefore, the work is ready for publication.

Although some limits related to the work still remain, which include a potentially broader evaluation and a deeper analysis, this work offers an interesting perspective that deserves to be shared to the community.

**Audience:**

Yes

**Audience Explanation:**

The typical audience includes a mix of machine learners and computer visionists. This work attacks an actual problem, debiasing. This work is clearly of relevance for the community.

**Claims And Evidence:**

Yes

**Claims Explanation:**

Overall, this work proposes the use of known methods in a different setup, without making overly strong claims. The major weakness of this work is an incomplete theoretical analysis - besides the one provided as supplementary material - , which makes this work somewhat incomplete. However, given that the nature of this work is essentially empirical, it is provided sufficient evidence in this regard.